# Marine Cyanobacteria and Microalgae Metabolites—A Rich Source of Potential Anticancer Drugs

**DOI:** 10.3390/md18090476

**Published:** 2020-09-19

**Authors:** Arijit Mondal, Sankhadip Bose, Sabyasachi Banerjee, Jayanta Kumar Patra, Jai Malik, Sudip Kumar Mandal, Kaitlyn L. Kilpatrick, Gitishree Das, Rout George Kerry, Carmela Fimognari, Anupam Bishayee

**Affiliations:** 1Department of Pharmaceutical Chemistry, Bengal College of Pharmaceutical Technology, Dubrajpur 731 123, West Bengal, India; 2Department of Pharmacognosy, Bengal School of Technology, Chuchura 712 102, West Bengal, India; sankha.bose@gmail.com; 3Department of Phytochemistry, Gupta College of Technological Sciences, Asansol 713 301, West Bengal, India; sabyasachibanerjee04@gmail.com; 4Research Institute of Biotechnology and Medical Converged Science, Dongguk University-Seoul, Goyang-si 10326, Korea; jkpatra.cet@gmail.com (J.K.P.); gitishreedas@gmail.com (G.D.); 5Centre of Advanced Study, University Institute of Pharmaceutical Sciences, Punjab University, Chandigarh 160 014, Punjab, India; jmalik_pu@hotmail.com; 6Department of Pharmaceutical Chemistry, Dr. B.C. Roy College of Pharmacy and Allied Health Sciences, Durgapur 713 206, West Bengal, India; gotosudip79@gmail.com; 7Lake Erie College of Osteopathic Medicine, Bradenton, FL 34211, USA; kilpatrickkaitlyn@gmail.com; 8Post Graduate Department of Biotechnology, Utkal University, Bhubaneswar 751 004, Odisha, India; routgeorgekerry3@gmail.com; 9Department for Life Quality Studies, Alma Mater Studiorum-Università di Bologna, 47921 Rimini, Italy

**Keywords:** marine, microbes, microalgae, cancer, prevention, therapy, in vitro, in vivo, clinical studies

## Abstract

Cancer is at present one of the utmost deadly diseases worldwide. Past efforts in cancer research have focused on natural medicinal products. Over the past decades, a great deal of initiatives was invested towards isolating and identifying new marine metabolites via pharmaceutical companies, and research institutions in general. Secondary marine metabolites are looked at as a favorable source of potentially new pharmaceutically active compounds, having a vast structural diversity and diverse biological activities; therefore, this is an astonishing source of potentially new anticancer therapy. This review contains an extensive critical discussion on the potential of marine microbial compounds and marine microalgae metabolites as anticancer drugs, highlighting their chemical structure and exploring the underlying mechanisms of action. Current limitation, challenges, and future research pathways were also presented.

## 1. Introduction

According to a World Health Organization (WHO) report, by 2030 there will be 21 million new cases of cancer and 13 million deaths due to this disease [1]. Currently, 13% of all deaths worldwide are induced by cancer, and it is estimated that 30% of such deaths can be avoided by modifying or preventing the major risk factors, such as tobacco smoking, radiation exposure, alcohol, and infections [2]. Nearly all anticancer medications currently on the market have serious adverse effects, and therefore, new and safer anticancer drugs are desirable. Although there has been a decline in the interest of the pharma industry in natural products in the recent past, they are still the best platform for providing novel, effective, and unique chemical structures that may have considerable potential to treat or prevent cancer or serve as scaffolds or lead molecules for more effective anticancer drugs. From 1981 to 2010, approximately 1355 drugs were approved for therapeutic application, and among these, 128 were anticancer drugs with approximately 35% of them from either natural products or compounds extracted from natural products [3].

More than 70% of medicinal products for clinical use are derived from natural products, and this also extends into cancer chemotherapy, in which natural products make up most of the current chemotherapy medications [4]. Nature remains a rich resource of bioactive and varied chemotypes, and while comparatively very few of the extracted natural products are developed into clinically useful drugs by themselves, these specific molecules also act as preparatory models for more efficacious analogues and prodrugs through the implementation of chemical methodology. The main priority of natural products for the identification and development of new anticarcinogenic pharmaceutical drugs and the value of cross-disciplinary collaboration in the extraction of novel molecular functionalities from natural product resources have been extensively investigated [5].

However, the ocean holds a broad reservoir of marine species full of natural pharmaceutical components of potential significance. Marine bioprospecting is a recent phenomenon; thus, aquatic life is a largely unexplored field of research [6,7,8,9]. Terrestrial life is the source of most pharmaceutically effective natural products [10]. For centuries, marine flora has been exploited for their potential medicinal applications throughout the world [11]. Among the marine organisms, bacteria, fungi, actinobacteria, seaweeds, and sponges have been utilized for cancer treatment [12,13,14,15,16].The most effective anti-cancer drugs are natural products. However, the natural product research and innovation phases are expensive, tedious, and time-consuming [17,18].

Few marine pharmaceutical products for marketing have been licensed and some molecules are being clinically trialed within Phases I and III, along with preclinical studies. It is notable that among the scientifically developed marine compounds, four compounds are used in cancer treatment, such as cytarabine (Cytosar), trabectedin (Yondelis), eribulin mesylate (Halaven) and the conjugated antibody brentuximab vedotin (Acentris).

The dynamic and extremely sensitive global marine ecosystem covers about 70.8% of Earth [19]. This extreme biodiversity encompasses a heterogeneous array of micro- and macro-organisms. Among them, microorganisms like marine bacteria, fungi, and micro-algae orchestrate a pivotal role in restoring the balance within the aquatic environment by being both a producer and decomposer [20]. Marine microalgae are essential ecologically which are used as food and medicinal products from ancient times. Marine microalgae are eukaryotic plants that contribute to drug discovery by their metabolicplasticity, which can trigger the production of several compounds with possible applications in combating various diseases, such as cancer [21]. Presently, these aquatic species have gained increasing acclaim for their bioactive metabolites, which provides an unparalleled potential for a range of pharmaceutical activity. The distribution of marine species differs depending on the type of open sea ecosystem, such as pelagic environment, epipelagic environment, mesopelagic zone, bathypelagic zone, abyssopelagic zone, and benthic environment. These diversified divisions of the aquatic world harbors heterogeneous species of marine microflora and microalgae. Therefore, summing all the types of microflora and microalgae, and their extracted bioactive chemical metabolites, would be too tedious to elaborate in a single article. Thus, only certain specific microflora including bacteria, fungus, and microalgae (cyanobacteria), along with their potent bioactive metabolites displaying anticancer activity are emphasized. The most common bioactive compounds having anticancer activity from these marine sources are alkaloids (staurosporine, ambigols, amycolactam, and marinoquinoline A), polyketides (chromone, engyodontiumones H, pestalpolyol I, hytidchromone A, B, C, and E), terpenes (meroterpenes, diterpene, and scopararane I), peptides (beauvericin, polymyxin B and other non-ribosomal peptides), nucleosides (cytarabine, gemcitabine and other nucleoside analogues) and carbohydrates (laminaran, alginic acid, and other sulfated polysaccarides) [22,23,24].

Microflora and, up to certain extent, microalgae could be regarded as chief drivers of nutrient transformations in a marine ecosystem. Shift in abiotic factors of marine ecosystem, such as temperature, salinity, nutrients, oxygen, solar energy, water clarity, tides, waves, aerial exposure and current has immensely influenced the production and secretion of marine metabolites/ bioactive chemical compounds from these organisms. To survive under these harsh abiotic factors, the marine microflora tends to form a symbiotic relationship with other marine microorganisms. These synergetic relationships enable them to endure and secrete a number of exotic secondary metabolites/bioactive compounds which they would not do under natural circumstances [25]. Over the last two decades, natural product research has observed a paradigm shift from terrestrial areas to oceans. Approximately 15,000 marine-origin metabolites, which have demonstrated cytotoxicity in cancer models, have been identified during the past three decades and from these, 28 agents are under clinical trials for their anticancer properties [26]. There is myriad of structural motifs undiscovered from the marine ecosystem and these metabolites can serve as new potent anticancer agents.

There are only a few previous reviews that present an in-depth overview of this important field of research. Many of the preceding publications focus exclusively on the compilation of marine secondary metabolites for study on natural products [27,28,29,30,31]. This review looks at the pharmacology of marine species with respect to anticancer drug molecules which have shown significant bioactivity to become drug or is in queue to enter clinical trials. The data procured covers published preclinical study, research papers and review of marine molecules isolated from a diverse group of marine algae, and cyanobacteria. This review evaluates the anticancer effects of numerous marine cyanobacterium and microalgae secondary metabolites, emphasizing on their chemical structures and highlighting the mechanisms of action that underlie their pharmacological activities.

## 2. Literature Search Methodology

In vitro, in vivo and clinical research investigating the anticancer potential of secondary secreted metabolites of marine species by modulating various pathways were screened utilizing credible repositories which include PubMed, ScienceDirect, Web of Science, SpringerLink, Scopus, and Google Scholar. Comprehensive papers released in peer-reviewed publications up to April 2020 have been included. There was no time restrainst on publication year. Only publications written in English have been listed and used in this article. The exclusion criteria for not setecting articles included non-English language publications, letters to editors, conference abstracts, and unpublisged reports. The keywords included within the literature quest are cancer, tumor, proliferation, cytotoxicity, apoptosis, marine, microbes, microalgae, cyanobacteria, prevention, therapy, In vitro, in vivo, and clinical trials. The bibliography of selected primary literature was also searched for additional relevant papers.

## 3. Various Classes of Secondary Metabolites of Marine Cyanobacterium and Microalgae

### 3.1. Alkaloids

Alkaloids are naturally occurring/synthetic organic compounds collectively used to describe the diverse groups of heterocyclic compounds having alkali-like properties and having at least one single nitrogen atom within its structure [32,33]. Presently, these nitrogen atoms containing heterocyclic compounds could be classified based on the basis of carbon skeleton resemblance contained in biochemical precursors such as ornithine, lysine, tyrosine, and tryptophan, having either indole and isoquinoline, or pyridine moieties [34]. Marine alkaloids could also be grouped into indoles, halogenated indoles, and phenylethylamines [35]. Marinoquinoline A is an anticancer alkaloid produced by *Catalinimonas alkaloidigena*, a marine bacterium [36] along with 13 other alkaloid metabolites. *Pseudoalteromonas tunicata* and *P*. *citrea* are two marine bacteria that secrete a yellow-pigmented alkaloid that belongs to a group called tambjamines, and this showed anti-tumor activity along with antimicrobial, antifungal, and antimalarial activity [37,38]. Calothrixins A and B are alkaloids containing a phenanthridine moiety that are isolated from *Calothrix sp*. Both the alkaloids showed significant cytotoxicity to human cervical carcinoma cells (HeLa) [39]. Demay et al. [40] have comprehensively reviewed a variety of bioactive metabolites of cyanobacteria, such as *Hapalosiphon fontinalis*, *Fischerella musicola*, *F. ambigua*, *H*. *welwitschii*, and *Westiella intricate*, as among which hapalindole-like alkaloid, and ambigols, possess cytotoxic activity.

### 3.2. Polyketides

Polyketides are a broad group of compounds which are biosynthesized as analogs generated by a sequence of modular enzymes act as biocatalysts from precursors which contain alternating carbonyl and methylene groups(-CO-CH_2_-) [41]. Then the compound undergoes decarboxylative condensation and modification of the acetate or the propionate chains primarily by reduction reaction, dehydration reaction, cyclization reaction, and aromatization reactions. *Streptomyces koyangensis*, a marine bacterium that produces two abyssomicins which have antitumor activity, but in-depth investigation is scanty [42]. Extracts of marine cyanobacteria like *Nostoc spongiaeforme* and *N. linckia*, contained a polyketide called borophycin that showed the strong anticancerous effect against human colon carcinoma cell lines (LoVo) [43].

### 3.3. Terpenes

Terpenes are the hydrocarbon compounds formed from 5-carbon isoprene units assembled to generate a vast range of skeletons, which are used by various enzymes to conjugate functionality and alter oxidation. These cyclic molecules can be categorized as monoterpenes, diterpenes, triterpenes (steroids), tetraterpenes (carotenoids), sesquiterpenes, and sesterterpenes based on the isoprene units it contains [44]. Presently it is acknowledged that marine microflora is an excellent source of these terpenes, and among which few terpenes exhibited their anticancer activity apart from other biological properties [22].

### 3.4. Peptides

Peptides are the definite protein fragments which provide optimistic impacts on human health [41,42,43,44,45]. Peptides are usually inert within the parent protein chain and may exhibit several physiological tasks upon proteolysis [46,47]. Enzymatic hydrolysis has played a significant role in the synthesis of peptide compounds in marine species [48,49]. Numerous prospective reports were documented about the utilization of marine peptides for pharmaceutical developments, including antitumor properties [50,51]. Cyclic and linear peptides have been established as potentially effective cytotoxic agents. These peptides possess cytotoxic, antimicrobial, specific ion channel-blocking, and other pharmacological activities with innovative chemical structures correlated with the actual mode of action [50].

A new peptide, polydiscamide A, and its analogs possess anti-tumor activity [50]. A certain number of marine peptides were successfully evaluated through clinical research and have now been available as formulated drugs in the market under various trade names.

A cyclic depsipeptide known as Apratoxin A exhibited effective cytotoxic potential against the human cervical carcinoma cells (HeLa) by triggering cell cycle arrest [52]. Peptides extracted from *Lyngbya* sp. and *Nostoc* sp., were reported to possess promising anticancer activity by the disruption of microfilaments, inhibition of secretory pathways, and influencing other intracellular pathways [26].

### 3.5. Nucleosides

Nucleosides belong to a class of organic compounds and are commonly known as the nitrogen glycosides of purines and pyrimidines; with their phosphate esters, they are called nucleotides [53,54]. These are the important constituents of all living cells and are associated with various fundamental physiological processes [53,54]. Marine microbes are skilled at manufacturing different nucleosides forms with unfamiliar structures and associated biological properties [53,54,55,56,57]. A few of these nucleosides with substantial pharmacological activities have been documented [56,57,58]. The potent biological properties of marine-derived nucleosides have stimulated the production of some analogs [59,60,61]. Marine nucleosides themselves have exhibited numerous bioactive potentials including the anticancer, antiviral, muscle relaxant, hypertensive and vasodilator activities [54].

### 3.6. Carbohydrates

Carbohydrates form the major component of aquatic organisms and are considered the significant food source of these organisms, particularly algae [62]. These compounds vary extremely in their molecular structure and resemble a pool of sulfated polysaccharides manufactured fixation of sulfur and carbon by the photosynthetic organisms [62]. Carbohydrates are classified by source in three groups: plant polysaccharides [63], animal polysaccharides [64], and microbial polysaccharides [65], which include both neutral and negatively charged saccharides with varying size [24]. Examples include either the nitrogen-linked or oxygen-linked oligosaccharides in glycoproteins, glycosaminoglycan in proteoglycans, glycolipids, sulfated fucans, and sulfated galactans [24,66]. However, their chemical compositions and arrangements are complex and heterogeneous in nature [67].

Marine carbohydrates are produced by a specific enzymatic hydrolysis process of polysaccharides [67]. The enzymatic breakdown of the sulfated polysaccharides involves a set of enzymes that can cleave the glycosidic bond and eliminate the sulfate groups from the carbohydrate backbone [62]. These marine-derived carbohydrates, including alginic acid, agar, carrageenan, chitin, cellulose, chitosan, fucan, glucan, glucosamine glycan, and laminaran; these possesses a wide number of substantial bioactive properties which includes the anticancer potentials [11,67,68]. These carbohydrate-based compounds exhibited anticancer effects against a number of carcinoma cells by modulating the innate immune system, which triggered the chemotactic response of macrophages and natural killer cells to the target location and have them produce their tumoricidal cytokines [12]. Fucoidan is a sulfated polysaccharide present as a metabolite in the brown algae cell wall and has been shown to inhibit atherosclerosis, angiogenesis, and metastasis [69] in the human lymphoma cell line (HS-Sultan) by subsequent activation of caspase-3, and downregulating kinase activity [12,70].

## 4. Secondary Metabolites of Marine Cyanobacteria and Microalgae at Various Phases of Clinical Research

Certain anticancer compounds from marine cyanobacteria and microalgae are currently undergoing clinical trials [12,22]. These natural bioactive compounds exhibited anticancer activity by regulating macromolecule expression induced in cancer cells via oncogenic signal transduction pathways [71].

Fewer than 10% of marine pharmacologically active compounds have been tested against diverse cancer types [60]. For example, the compounds dolastatin 10, ET-743, and bryostatin 1 are tested and analyzed in clinical research [12]. A variety of dolastatins and associated molecules were extracted from filamentous cyanobacteria of the genera Symploca and Lyngbya [72].These are small oligopeptides containing four unique non-protein amino acids—dolaphenine, dolaproline, dolaisoleucine, and dolavaline. Dolastatin 10 is a linear peptide while dolastatin 15 is a seven-unit depsipeptide agent, and both are potent cytostatic peptides that arrest cell division [59]. This was proven by studies that showed that dolastatins, especially dolastatin 10, attached to tubulin at the guanosine triphosphate position, leading to the disruption of its normal function and triggering metaphase cell cycle arrest [73]. Dolastatin 10 reached Phase I of clinical research in 1990s and has advanced to Phase II of clinical research [74]. However, because it developed peripheral neuropathy in over 40% of patients, it was discontinued [74]. Even so, it was the basis for more efficient derivatives to come into existence. One of its derivatives, the monoclonal antibody-drug conjugate *Brentuximab vedotin* (Adcetris) has been licensed for its anticancer activity against Hodgkin’s lymphoma [75].

Various other derivatives as antibody-drug conjugates (*Polatuzumab vedotin*, *Depatuxizumab vedotin*) are under Phase III of clinical research; *Enfortumab vedotin*, *Glembatumumab vedotin*, *Tisotumab vedotin*, and others are under Phase II of clinical trials, whereas ABBV-085, ASG-15ME, and AGS-67E are under Phase I of clinical research for different types of cancers (www.clinicaltrials.gov). Soblidotin (TZT-1027) is a synthetic analog of dolastatin 10 that is more potent against cancer than its parent compound and other established anticancer medications such as podophyllotoxin and vincristine [76]. In addition to inhibiting tubulin function, TZT-1027, a vascular disrupting agent, causes a collapse in the vasculature of the tumor, causing cell death [77]. After completing Phase I and II trials, TZT-1027 is in the Phase III of clinical research under Aska Pharmaceuticals [78].

Synthadotin (ILX-651) is a synthetic pentapeptide derivative of dolastatin 15, is a potent antitumor agent in patients having metastatic melanoma in advanced stage [75]. It has been shown to inhibit microtubule nucleation [79]. Other dolastatins showed cardiovascular toxicity, but ILX-651 has not exhibited such toxicity [79]. It has successfully completed both Phase I and Phase II of clinical research and was found to be well tolerated and completely safe [80]. Another compound, bryostatin 1, has successfully completed Phase II of clinical research for the treatment of melanoma, non-Hodgkin’s lymphoma, renal cancer, and colorectal cancer [12]. The marine bioactive compounds play a significant major role in the discovery of anticancer drugs, and these are classified as the antimigration, antimetastatic, anti-invasion, antitubulin agents and growth inhibitors, apoptosis inducers, autophagy, and antiangiogenic agents [22]. Also, because of their importance in signal transduction pathways, an additional family that includes proliferation inhibitors of mitogen-activated protein kinases have also been incorporated [22].

Salinosporamide Ais a cytotoxic bicyclic β-lactone-γ-lactam isolated from the *Salinispora tropica,* amarine actinobacterium [81]. It is an inhibitor of proteasome where the halogenation in its side chain containing ethyl functional group helps to irreversibly bind the 20S proteasome [81]. By binding to this enzyme, salinosporamide A triggers apoptosis of cancer cells [82]. Phase I and Phase II clinical research have been successfully completed for the treatment of solid tumors lymphoma and multiple myeloma [83]. The finding of the Phase 1 trial demonstrated good anticancer activity among multiple myeloma patients, with decent health and a non-cross-reactive toxicity profile [83]. Salinosporamide A did not induce peripheral neuropathy, thrombocytopenia, or myelosuppression, which were observed with other proteasome inhibitors [83]. Similarly, Phase II findings showed enhanced therapeutic function and increased duration of the inhibition of the proteasome [75]. Both the United States Food and Drug Administration (US-FDA) and European Medicines Agency (EMA) declared it to be an orphan medicine targeting multiple myeloma [75]. At present salinosporamide is undergoing Phase III studies for newly diagnosed cases of glioblastoma [75].

## 5. Marine Cyanobacteria Metabolites with Anticancer Property

Owing to their extraordinary abundance, marine cyanobacteria have drawn tremendous interest within the marine ecosystem. Some of the anticancer agents from marine cyanobacteria as shown in Table 1 are discussed below.

### 5.1. Anthracyclines

Komodoquinone A (**1**) (Figure 1), produced by *Streptomyces* sp. KS3. This is a new anthracycline that caused neurotogenesis (process of forming new neurites) in the neuro 2A neuroblastoma cell line [84].

### 5.2. Phenoxazin-3-One Compounds

The novel anticancer antibiotics chandrananimycins A (**2**), B (**3**), and C (**4**) with phenoxazin-3-one moiety have been isolated from the marine *Actinomadura* sp. [85]. These compounds exhibited anticancer activity by inhibiting the proliferation of cancer cell lines such as, CCL HT29 (colon cancer cell), MEXF 514L (melanoma cells), LXFA 526L, LXFL 529L (lung cancer cells), CNCL SF268, LCL H460, MACL MCF-7 (breast cancer cells), and PRCL PC3M, RXF 631L (kidney tumor) [85].

### 5.3. Polyketides

Ankaraholide A (**5**) is a glycosylated swinholide compound, which is procured from *Geitlerinema* sp. [86]. It inhibited the proliferation of NCI-H460, Neuro-2a, and MDAMB-435 cells [96]. Swinholide A (**6**) was initially obtained from the marine sponge *Theonella swinhoei*. Later, it was also reported to be the metabolites of the marine cyanobacterium *Symploca* sp. It exhibited its antitumor activity by disruption of actin [86].

### 5.4. Peptides

Symplostatin 1 (**7**) (Figure 1) is an analogue of dolastatin 10, isolated from *Symploca hydnoides*, a marine bacterium. The antimitotic activity of symplostatin was shown against a panel of cancer cell lines, such as MDA-MB-435, SK-OV-3, NCI/ADR, A-10, and HUVEC [87].It also showed profound antitumor activity against murine colon 38 and murine mammary 16/C carcinoma cells. This metabolite triggered the phosphorylation of Bcl-2, micronuclei formation, caspase-3 activation, and induced apoptosis that led to cell cycle arrest at the G2/M Phase. It also prevented the accumulation of tubulin [87]. Grassypeptolideis a macrocyclic depsipeptide formed by the cynobacteria *Lyngbya confervoides* [88]. It showed cytotoxic antiproliferative activity against various cell lines, such as human osteosarcoma (U2OS), cervical carcinoma (HeLa), colorectal adenocarcinoma (HT29), and neuroblastoma (IMR-32) cell lines [88]. Grassypeptolide A (**8**) (Figure 1), B (**9**) and C (**10**) (Figure 2), significantly inhibited the proliferation of colorectal adenocarcinoma (HT29) cell lines and cervical carcinoma (HeLa) cell lines in a concentration-dependent manner by inducing cell cycle arrest at either G1 the Phase or G2/M Phase [89].

Curacin A (**11**) (Figure 2), a linear and complex hybrideketopeptide, was the first curacin that was isolated from extracts of the Caribbean cyanobacterium *Lyngbya majuscule* [90]. After curacin A, other curacin compounds, namely curacin B, C, and D, were also identified as the constituents of *L. majuscule* [91]. Amongst these, curacin A was the most active anticancer compound that inhibited the proliferation of non-small cell lung cancer cells (A549) by triggering apoptosis and causing cell cycle arrests in the G2-M Phase [92]. It acts by binding to tubulin at colchicine binding site and acting as a competitive antagonist, and tubulin polymerization inhibitor [93]. Its structural activity relationship (SAR) studies indicated that the existence of four double bonds, a conjugated diene, a readily oxidized thiazoline heterocycle, and high lipophilicity are the factors that govern the pharmacological function of curacin A [94]. Various analogues were prepared, but they exhibit activity, lower than that the parent compound [95].

Tasiamide B (**12**) (Figure 2), a linear peptide extracted from cyanobacteria *Symploca* sp., demonstrated potent toxicity against human nasopharyngeal carcinoma (ĸB) and LoVo cancer cells, respectively [96]. Apratoxins (Figure 2) are cytotoxic cyclic depsipeptides that have a novel polyketide and peptide fragmented structure [98]. Apratoxin A (**13**) (Figure 2) was found in an aquatic cyanobacteria *Lyngbya majuscula*. It disrupted the secretory pathway of U2OS osteosarcoma cells and even induced arrest at G1 Phase of the cell cycle in HeLa cervical carcinoma. It also demonstrated significant cytotoxic behavior in human tumor cell lines, such as LoVo cells and epidermal ĸB carcinoma cancer cells, respectively [97]. Apratoxin A inhibited the translocation of proteins by specifically attacking a central subunit of the protein translocation receptor, Sec61α [99]. Apratoxin B (**14**) and C (**15**) (Figure 2), extracted from *Lyngbya* sp., has high cytotoxicity against ĸB oral epidermoid cancer cells and LoVo colon cancer cells [97]. Apratoxin D (**16**) (Figure 2), collected from *Lyngbya majuscule* and *Lyngbya sordid*, demonstrated significant cytotoxicity against the H-460 lung cancer cell line [98]. Apratoxin E (**17**) (Figure 3) extracted from *Lyngbya bouilloni* demonstrated strong antiproliferative action against diverse cancer cell lines, such as U2OS osteosarcoma, HT29 colon adenocarcinoma, and HeLa epithelial carcinoma [99].

Apratoxin F (**18**) and G (**19**) (Figure 3), containing N-methyl alanine in their composition, has been reported to be isolated from *Lyngbya bouilloni* and have strong cytotoxicity against H-460 lung cancer and HCT-116 colorectal cell lines [100]. Cyclic depsipeptidesaurilide B (**20**) and C (**21**) (Figure 3), isolated from *Lyngbya majusculus*, demonstrated cytotoxicity in human lung tumor cell line NCIH460 and neuro-2a mouse neuroblastoma cells [101]. Aurilides binds prohibitin 1 (PHB1) in the mitochondria, stimulates optic atrophy 1 (OPA1) proteolytic synthesis and contributes to mitochondrial apoptosis [102].Coibamade A (**22**) (Figure 3) is a cyclic depsipeptide extracted from marine cyanobacterium *Leptolyngbya* sp. It showed strong cytotoxic activity against a triple-negative breast cancer cell line (MDA-MB-231) [103]. The antiproliferative activity of the active metabolite is associated with the cell cycle arrest at the G1 Phase. Similar observations were reported, where glioblastoma cell lines, such as U87-MG and SF-295, were treated with coibamide A. It also inhibited the proliferation of HUVECs by inducing cell morphology change and reducing the expression level of vascular endothelial growth factor receptor 2 (VEGFR2) [104]. Cobamide A induced autophagy associated cell death of human U87-MG glioblastoma cells and SF-295 glioblastoma cells and mouse embryonic fibroblasts [105].

The structurally distinct cyclic depsipeptideshoiamide A (**23**) and B (**24**) (Figure 4), isolatedfrom *Lyngbya majuscule* and *Phormidium gracile*, exhibited strong cytotoxicity [106]. Hoiamide A displayed moderate cytotoxicity against mouse neuroblastoma (neuro-2a) cells and human lung adenocarcinoma (H460) cells, while hoiamide B exhibited weak cytotoxicity against H460 and no inhibition against neuro-2a cells [106].Homodolastatin 16 (**25**) (Figure 4), a cyclic marine depsipeptide isolated from *Lyngbya majuscule*, appears to have a modest cytotoxic effect against esophageal (WHCO1 and WHCO6) cell lines and cervical cell line (ME180), respectively [107]. Largazole (**26**) (Figure 4), a cyclodepsipeptide isolated from *Symploca* sp., has demonstrated a significant hindrance in the development of extremely invasive transformed human mammary epithelial cells (MDA-MB-231) in a concentration-dependent manner. The growth of HT29 colon and IMR-32 neuroblastoma cells has also been significantly inhibited. Similar pharmacological activity was observed towards transformed fibroblastic osteosarcoma (U2OS) cells [108]. It causes cell-cycle arrest colon cancer cell line HT29 at the G2/M Phase [110]. It is an inhibitor of histone deacetylase (HDAC) in vivo in the tumor xenograft model of HCT116 [109].

Lyngbyabellin A (**27**) and B (**28**) (Figure 4) are cyclic depsipeptides extracted from *Lyngbya majusculus* which possessed a strong actin polymerization mechanism with significant cytotoxicity against ĸB and LoVo cells [110]. Lyngbyabellin E–I (**29**–**33**) (Figure 5) demonstrated cytotoxicity to NCI-H460 human lung tumor and neuro-2a mouse neuroblastoma cells [110,111]. Lyngbyabellin N (**34**) (Figure 5) extracted from *Moorea bouillonii* demonstrated cytotoxicity in human lung carcinoma (H-460) and colon cancer cell lines (HCT116) [112]. Majusculamide C (**35**) and desmethoxymajusculamide C (**36**) (Figure 6) are cyclic depsipeptides derived from the *Lyngbya majuscule*, a marine cyanobacterium. Majusculamide C was identified as cytotoxic and showed strong activity against several cancer cell lines, such as ovarian carcinoma (OVCAR-3), kidney cancer (A498), lung cancer (NCI-H460), colorectal cancer (KM20L2), and glioblastoma SF-295 cell lines [113]. Once screened against HCT-116 human colon carcinoma cells, desmethoxymajusculamide C has a good and a selective antitumor effect [110].

Obyanamide (**37**) (Figure 6), a cyclic depsipeptide isolated from *lyngbya confervoides*, has demonstrated significant cytotoxicity to the ĸB and LoVo cells [114]. Palau’amide (**38**) (Figure 6) is a cyclical depsipeptide isolated from the same marine cyanobacteria *Lyngbya* sp., which has shown significant cytotoxicity to ĸB cells [115]. Palmyramide A (**39**) contributes to cytotoxicity in neuro-2a cells possibly through blocking the voltage regulated sodium channel. Palmyramide A has shown moderate cytotoxic effects on human lung cell line H-460 [116]. Pitipeptolides A (**40**), and B (**41**) (Figure 6), cyclic depsipeptides isolated from a marine cyanobacterium *Lyngbya majuscule*, were reported to possess cytotoxic action against HT29 colon adenocarcinoma cancer cells [117]. Apart from that pitipeptolide A and B also exhibit cytotoxicity against LoVo cells [118].

Pitiprolamide (**42**) (Figure 7) a cyclic depsipeptide obtained from *Lyngbya majuscule*, showed cytotoxicity against MCF7 breast adenocarcinoma and HCT116 colorectal carcinoma cell lines [119]. Tasipeptins A (**43**) and B (**44**) (Figure 7), cyclic depsipeptides derived from *Symploca* sp., exhibitedcytotoxicity against ĸB oral epidermoid cancer cells [120]. Ulongapeptin (**45**) (Figure 7), a cyclic depsipeptide obtained from *Lyngbya* sp., possessed cytotoxicity against ĸB oral epidermoid cancer cells [121]. Veraguamides A–G (**46**–**52**) (Figure 7 and Figure 8) are cyclic hexadepsipeptide obtained from *Symploca* cf. *hydnoides* [120]. These metabolites demonstrated cytotoxic property against HT29 colon adenocarcinoma and HeLa cervical carcinoma cells [122]. In addition, it also possessed potent cytotoxicity against the H-460 human lung cancer cell line [123]. Wewakpeptins A–D (**53**–**56**) (Figure 8 and Figure 9) are depsipeptides obtained from *Lyngbya semiplena* that have exhibited anticancer activity by inhibiting the proliferation of H-460 lung cancer cells [124].

Nostocyclopeptide A1 (**57**) and A2 (**58**) (Figure 9), are cyclic heptapeptides isolated from *Nostoc* sp., displayed cytotoxicity against ĸB oral epidermoid cancer and LoVo colon cancer cells [125]. Symplocamide A (**59**), isolated from *Symploca* sp., is a cyclopeptide that showed potent cytotoxicity to H-460 non-small cell lung cancer cells and neuro-2a neuroblastoma cells [110]. Tasiamide (**60**) (Figure 9) is a cyclopeptide isolated from the cyanobacterium *Symploca* sp., which showed strong cytotoxicity against human nasopharyngeal carcinoma (ĸB) and LoVo cells [126]. Belamide A (**61**) (Figure 9) is categorized as a linear tetrapeptide with structural similarities to dolastatin 10 and dolastatin 15 that is isolated from cyanobacteria *Symploca* sp. It exhibited good cytotoxicity against HCT-116 colon cancer cell line [127]. This has been used to destabilize tubulin in vascular smooth muscle cells (A10) to induce antimitotic action [127].

Bisebromoamide (**62**) (Figure 10) a peptide that is a marine toxic substance isolated from *Lyngbya* sp. [128,129]. It exhibited inhibition of the phosphorylation of extracellular signal-regulated kinases (ERK) in normal rat kidney epithelial (NRK) cells stimulated by platelet-derived growth factor [129]. Dragonamide (**63**) and pseudodysidenin (**64**) (Figure 10) are lipopeptides isolated from *Lyngbya majuscule*, and both exhibited cytotoxicity against P-388, A-549, HT-29, and MEL-28 carcinoma cells [130]. Kalkitoxin (**65**) (Figure 10), another lipopeptide is isolated from *Lyngbya majuscule*, was shown to decrease HCT116 colon cancer cell survival [131]. Somocystinamide A (**66**) (Figure 10), a lipopeptide obtained from the same cyanobacterium, exhibited antiproliferative activity against Jurkat (T cell leukemia), CEM leukemia, A549 lung carcinoma, Molt4T leukemia, M21 melanoma and U266 myeloma cells. It induced apoptosis via activation of caspase 8 [132].

Malyngamide 2 (**67**) (Figure 10), isolated from *Lyngbya majuscule* and *Lyngbya sordid*, displayed cytotoxicity against H-460 human lung carcinoma cells [133]. Malyngamide C (**68**), J (**69**), and K (**70**) (Figure 10) are documented to be the metabolites of *Lyngbya majuscule*, exhibited cytotoxicity against some cancer cell lines [134]. Malyngamide C showed cytotoxicity against NCI-H460, neuro-2a, and HCT-116 cancer cell lines [134]. Malyngamide J and K displayed cytotoxicity against NCI-H460 and neuro-2a cell lines [134].

Malevamide D (**71**) (Figure 11), a peptide ester isolated from *Symploca hydnoides*, exhibited strong cytotoxicity against a panel of cancer cell line, such as P-388, human lung carcinoma (A-549), and human colon carcinoma (HT-29), and human melanoma (MEL-28) cells [135].Malyngolide dimer (**72**) (Figure 11) is a cyclodepsipeptide obtained from *Lyngbya majuscule* exhibited cytotoxicity when evaluated against the H-460 human lung cell line [136].

Cryptophycin 1 (**73**) and cryptophycin 52 (**74**) (Figure 11) are macrolide depsipeptides that are potent cytotoxic molecules. These are microtubule inhibitors bearing similar mechanism of action as that of Vinca alkaloids. Cryptophycin 1 is isolated from *Nostoc* sp., (blue green algae/cyanobacterium), showed anticancer activity against L1210 murine leukemia cells [137]. It binds with tubulin and triggers the disruption of microtubule assembly [138]. It exhibited cytotoxic function against KB cells and LoVo cell lines by inducing apoptosis [139]. It also displayed antiproliferative activity against MDA-MB-435 mammary adenocarcinoma and SKOV3 ovarian carcinoma cell lines by triggering cell cycle arrest at the G2/M Phase [140]. Cryptophycin 52 is a synthetic derivative of cryptophycin. Interestingly, even though it reached Phase II of the clinical trials, its development has been halted due to severe side effects [141]. Lagunamide A (**75**) and B (**76**) (Figure 11) are cyclic depsipeptides extracted from the marine cyanobacterium *Lyngbya majuscule*. Both the metabolites exhibited potent cytotoxicity against P388 (a murine leukemia cell line) [142]. Lagunamide C (**77**) (Figure 11) showed its cytotoxicity against carcinoma cell lines, such as P388, A549, PC3, HCT8, and SK-OV3 [143].

### 5.5. Macrolides

Biselyngbyaside (**78**) (Figure 12), obtained from the *Lyngbya* sp., is a secreted macrolide glycoside that induced apoptosis by nuclear condensation of mature osteoclasts [144]. Cytotoxicity has been demonstrated against HeLa S3, SNB-78 (central nervous system cancer) and NCI H522 (lung cancer) cells [144]. Biselyngbyasid B (**79**) (Figure 12), obtained from *Lyngbya* sp., showed cytotoxicity against HeLa S3 cells and HL60 cells [145]. Biselyngbyaside E (**80**) and F (**81**) (Figure 12) showed cytotoxic activity against HeLa and HL60 cells [146]. Lyngbyaloside B (**82**) (Figure 12), a macrolide glycoside isolated from *Lyngbya* sp., was shown to have a cytotoxic impact on ĸB cells and had a slightly reduced effect on LoVo cells [147]. Further, 2-epi-lyngbyalosid (**83**) (Figure 12), a macrolide glycoside extracted from *Lyngbya bouillonii*, demonstrated cytotoxic properties against HT29 colorectal adenocarcinoma and HeLa cells [148].

In addition, 18E-lyngbyaloside C (**84**) and 18Z-lyngbyaloside C (**85**) (Figure 13) also displayed potent cytotoxic property against HT29 colorectal adenocarcinoma and HeLa cells [148]. Biselyngbyolide A (**86**) and B (**87**) (Figure 13) are macrolide metabolites that have been isolated from the marine cyanobacterium *Lyngbya* sp., and have shown strong cytotoxicity against HeLa S3 cells and HL60 cells [149].Koshikalide (**88**) (Figure 13), a macrolide isolated from *Lyngbya* sp., displayed cytotoxic activity against HeLa S_3_ cells [150]. Acutiphycin (**89**) and 20,21-didehydroacutiphycin (**90**) (Figure 13), both macrolides isolated from *Oscillatoria acutissima*, demonstrated cytotoxic activity against ĸB cells and NIH/3T3 cells [151]. Lyngbouilloside (**91**) (Figure 14), a macrolide glycoside isolated from cyanobacteria *Lyngbya bouillonii* showed cytotoxic effect against neuroblastoma cells [152]. Polycavernoside D (**92**) (Figure 14), another glycosidic macrolide isolated from cyanobacteria *Okeania* sp., exhibited cytotoxic effect against human lung cancer cells (H-460) [153].

### 5.6. Lactones

Tolytoxin (6-hydroxy-7-*O*-methyl-scytophycin B, **93**) (Figure 14), a macrolide isolated from the lyophilized cells of *Seytonema ocellaturn*, exhibited cell growth inhibition of a panel of mammalian cells [154]. Scytophycin A–E (**94–98**) (Figure 14 and Figure 15), were reported to be isolated from blue green alga *Seytonema pseudohofmanni*, exhibited cytotoxic effect against human carcinoma of nasopharynx cell (ĸB cells) [155]. Caylobolide A (**99**) (Figure 15) is a macrolactone obtained from cyanobacteria *Lyngbya majuscula* that has exhibited In vitro cytotoxicity against human colon tumor cells HCT 116 [156]. Caylobolide B (**100**) (Figure 15) is obtained from *Phormidium* sp., and this has exhibited cytotoxic activity against HT29 colorectal adenocarcinoma and HeLa cervical carcinoma cells [157].

### 5.7. Fatty Acid Amines

Isomalyngamide A (**101**) (Figure 16) belongs to the class of fatty acid amines, which is isolated from the cyanobacteria *Lyngbya majuscule* and has inhibited the proliferation ofbreast cancer MCF-7 and MDA-MB-231 cells [158]. Isomalyngamide A-1 (**102**) (Figure 16) is extracted from cyanobacteria *Lyngbya majuscule* and *Lyngbya sordida* and has been shown to inhibit the proliferation of MDA-MB-231 cells [158]. Jamaicamides A (**103**), B (**104**), and C (**105**) (Figure 16) are fatty acid amines containing compound extracted from *Lyngbya majuscule* and *Lyngbya sordida*, showed cytotoxicity to both the H-460 human lung and neuro-2a mouse neuroblastoma cell lines [159].

### 5.8. Pigment

Scytonemin (**106**) (Figure 17) is a yellow-green pigment obtained from *stigonema species* of blue-green algae (cyanobacteria). It showed antiproliferative activity against Jurkat T cells by inducing apoptosis. The formation of mitotic spindle and the protein serine/threonine kinase activity was inhibited by scytonemin [160,161].

### 5.9. Boron Containing Metabolite

Borophycin (**107**) (Figure 17) is a boron-containing metabolite derived from marine blue-green algae (cyanobacterium) *Nostoc spongiaeforme* and *N. linckia*. It effectively showed anticancer activity against human cancer cell lines, namelyκB colorectal adenocarcinoma and LoVohuman epidermoid carcinoma [48,162].

### 5.10. Phenanthridine Alkaloids

Calothrixins A (**108**) and B (**109**) (Figure 17) are phenanthridine alkaloids isolated from the marine cyanobacterium *Calothrix* sp. They possessed significant cytotoxicity and inhibited the proliferation of human carcinoma cell line (HeLa) [163,164]. It also inhibited the proliferation of CEM leukemia cells (human T-cell leukemia cells) by inducing cell cycle arrest at G1 and G2/M Phases [165].

## 6. Microalgae Metabolites as Anticancer Drugs with Their Mechanisms of Action

The following compounds have been reported from the microalgal species that have shown anticancer properties (Table 2).

### 6.1. Polyunsaturated Aldehydes (PUAs)

Polyunsaturated aldehydes (PUAs) are derived from the marine diatoms *Thalassiosira rotula*, *Skeletonema costatum*, *Phaeocystis pouchetii*, and *Pseudo-nitzschia delicatissima* [166]. The PUAs isolated from these diatoms are 2-trans-4-trans-decadienal (**110**), 2-trans-4-cis-7-cis-decatrienal (**111**), and 2-trans-4-trans-7-cis-decatrienal (**112**) (Figure 18) [167]. These compounds exhibited potent antiproliferative and cytotoxic activity on human colon adenocarcinoma cancer line Caco-2 [167]. Three more PUAs such as 2-*trans*-4-*trans*-heptadienal, 2-*trans*-4-*trans*–octadienal, and 2-*trans*-4-*trans*-7-octatrienal (octatrienal) were reported to be extracted from marine diatom *Skeletonema marinoi*, among which 2-*trans*-4-*trans*-octadienal (**113**), and 2-*trans*-4-*trans*-heptadienal (**114**) were reported to possess significant cytotoxicity against lung adenocarcinoma cell line A549 and colon (COLO 205) cancer cells [168]. These metabolites also induced apoptosis, which is evident from the chromatin condensation, loss of membrane integrity, and nuclear fragmentation. These PUAs induced cell cycle arrest of all the carcinoma cell lines at either the G1 or S Phase associated with upregulation of caspase-3 and apoptosis-inducing factor 1 (AIFM1) [168]. None of the PUAs showed any toxicity on the human non-tumorigenic lung epithelial cell line BEAS-2B [168].

### 6.2. Polysaccharide

Microalgae polysaccharides have shown bio-stimulant activity that has been proven to be effective for a number of industrial applications, although only a few studies have shown the potential to act as an anticancer agent.The potentiality of the polysaccharides varied with changes in molecular weight and sulfate content [169].

#### 6.2.1. Chrysolaminaran Polysaccharide

Chrysolaminaran polysaccharide (**115**) (Figure 18) is a polysaccharide which is isolated from diatom *Synedra acus*. It belongs to chrysolaminaran family. It showed antitumor activity on human colon cancer cell lines HTC-116 and DLD-1 by inhibiting cancer cell proliferation [170].

#### 6.2.2. Sulfated Polysaccharide

The sulfated polysaccharide was reported to be an important constituent of the brown seaweeds of *Undaria pinnatifida* and *Saccharina japonica* [171]. This has an antitumor function [171] and prevented the proliferation of human breast cancer (T-47D) and melanoma (SK-MEL-28) cell lines and prevents the colony development [172].

Fucoidans (**116**) (Figure 18), are sulfated polysaccharides isolated from the brown seaweeds of *Sargassum hornery, Eclonia cava,* and *Costaria costata* [173]. It showed antitumor activity against human skin melanoma cell line (SK-MEL-28) and human colon cancer cell line (DLD-1) [173]. It also has been documented that fucoidans are an active constituent of *Fucus evanescens* (a brown algae of the Okhotsk sea), exhibited in vivo antitumor and antimetastatic activity in C57Bl/6 mice, a preclinical animal model [174]. Low molecular weight fucoidan induced apoptosis through alteration of mitochondrial membrane potential by the release of cytochrome c, and inhibitions of Bcl-2, Bcl-xl, Mcl-1 antiapoptotic protein and also activated apoptosis-inducing factors, caspase-3, caspase-7, caspase-9, in MDA-MB-231 cells [175]. ERK1/2 pathway inhibition in human lung cancer cells (A549) led to antimetastatic effect imposed by fucoidan. It also inhibited the phosphoinositide 3-kinases/protein kinase B/ mechanistic target of rapamycin (PI3K/Akt/mTOR) pathway, associated with the downregulation of the expression levels of matrix metalloproteinase-2 (MMP-2) in the A549 human lung carcinoma cell line [176]. Fucoidan inhibited the phosphorylation of EGF receptor. Additionally, it also inhibited the phosphorylation of ERK, JNK, c-fos, and c-jun and activator protein-1 (AP-1) [177]. Over-sulfated fucoidan blocked the angiogenesis process by suppressing mitogenic and chemotactic response of the vascular endothelial growth factor (VEGF) [178]. In another reported study, fucoidan inhibited the proliferation of human hepatocellular carcinoma cells (Huh7) by downregulating the expression levels of chemotaxin CXCL12 and its receptor CXCR4 [179]. It also reduced the expression levels of transforming growth factor (TGF) receptor I and transforming growth factor receptor II proteins and controlled the associated signaling molecules of TGF and regulated the SMAD2, SMAD3, and SMAD4 protein phosphorylation. This could potentially be another type of novel revolutionary mechanism by which the fucoidans exhibited antitumor activity in the breast carcinoma cells by the influence of epithelial-mesenchymal transition [180].

#### 6.2.3. Alginic Acid

Alginic acid (**117**) (Figure 18), commonly known as algin, is an anionic polysaccharide obtained from the cell wall of brown algae or seaweeds *Sargassum wightii*. Nanoparticles containing alginic acid provided stimulating antitumor effect on H22 tumor-bearing mice [41]. This polysaccharide also binds with toxic substances and heavy metals that cause cancer present in the intestine and it exerted its activity by converting these toxic substances into non-toxic ones [181].

#### 6.2.4. Laminarin

Laminarin (**118**) (Figure 18) is a polysaccharide obtained from brown algae *Eisenia bicyclis* [182]. It exhibited anticancer activity by inhibiting the proliferation and inducing apoptosis, and cell cycle arrest at the subG1 Phase in ovarian clear cell carcinoma cells (ES2), and papillary serous adenocarcinoma (OV90) cell lines. PI3K/MAPK intracellular signaling mechanism is inhibited in ovarian cancer cells, as well as the increased release of cytochrome c associated with an increase in DNA fragmentation and expression level of apoptosis linked proteins. It also induced MMP loss in both the carcinoma cells, along with autophagy through the inactivation of ULK1 and P62 phosphorylation [183]. Similarly, laminarin and its sulfated analog displayed potential in vitro anticancer activity against JB6 Cl41 (normal mouse epidermal cells), and SK-MEL-28 (human malignant melanoma) cells. Inhibition of proliferation and migration of these cancer cells is associated with inhibition of MMP-2 and MMP-9 proteinases and down-regulation of ERK1/2 signaling mechanism [184]. Similarly, it also inhibited the colony formation of human colon cancer cell lines, such as HCT-116, HT-29, and DLD-1, and displayed cytotoxicity against various carcinoma cell lines [182,185,186,187,188]. Ji and Ji [189] reported the anticancer activity of laminarin and its sulfated analog against LoVo cells. It is often associated with induction of apoptosis, upregulation of the expression levels of death receptor 4 (DR4) and DR5, TNF-related apoptosis-inducing ligand (TRAIL), Fas-associated protein with death domain(FADD), Bid, tBid and Bax, and downregulation of pro-caspase-8, pro-caspase-3, and Bcl-2 [190,191]. Additionally, activation of casapse-8, casapse-3, casapse-6, casapse-7, and casapse-9 and increased release of cytochromec were observed following the treatment with laminarin and its analogs [190,191]. The involvement of the laminarin on the ErbB signaling mechanism indicatesanother mechanism of action behind its apoptosis induction in human colon cancer cell line (HT-29) associated with cell cycle arrest at subG1 and G2-M Phases [192].

### 6.3. Carotenoids

Carotenoids are tetraterpenoids, which are also classified as the pigments formed by plants, algae, bacteria, and fungi [193]. There are more than 1100 carotenoids that have been identified so far. The carotenoid’s general configuration is a polyene chain of 9–11 double bonds. This correlates with numerous pharmacological features, including anticancer behavior. Various xanthophylls carotenoids were identified, such as violaxanthin, siphonaxanthin, fucoxanthin, neoxanthin, zeaxanthin, lutein, and lactucaxanthin, to be the major constituents in microalgae [193,194,195].

#### 6.3.1. Violaxanthin

Violaxanthin (**119**) (Figure 19) is the active metabolite reported to be present in the dichloromethane extract of the green algae *Dunaliella tertiolecta* [196]. It induced early apoptosis associated with biochemical and morphological changes in MCF-7 cancer cell line, but it did not contribute to the fragmentation of DNA. Additionally, it also reversed the multidrug resistance (MDR) by inhibiting the P-glycoprotein (P-gp) and MRP1 in L1210 (human MDR1 gene-transfected mouse lymphoma cells) and MDA-MB-231 (human breast cancer cells) [197]. Similar observations were recorded where violaxanthin reversed the MDR in human MDR1 (gene-transfected mouse lymphoma) and MCF-7 (human breast cancer cell) [198].

#### 6.3.2. Neoxanthin

Neoxanthin (**120**) (Figure 19) is a xanthophyll carotenoid that possesses cytotoxic activity on HeLa and A549 cancer cells. It is even more cytotoxic than violaxanthin [199].

#### 6.3.3. Fucoxanthin

Fucoxanthin (**121**) (Figure 19) is a pigment belonging to the xanthophylls family and found in brown algae *Undaria pinnatifida* as a major carotenoid. It shows antiproliferative activity against human leukemia cell line (HL-60) by inducing apoptosis [200]. Various studies revealed the anticancer nature in which it inhibited proliferation by inducing apoptosis and cell cycle arrest at the G0/G1 Phase or G2/M Phase through various molecules and pathways involving Bcl-2 protein, MAPK, NF-κB, caspase-3, caspase-8, caspase-9, and GADD45 in which their expression levels were regulated by fucoxanthin [201]. Among various carotenoids, fucoxanthin has been thoroughly researched as an anticancer agent and it has been established as having a significant anticancer activity [200,202,203,204,205].

#### 6.3.4. Siphonaxanthin

Siphonoxanthin (**122**) (Figure 19) is a keto-carotenoid present as an active metabolite in green algae *Codium fragile*, *Caulerpa lentillifera*, and *Umbraulva japonica*. Siphonaxanthin demonstrated the anticancer effect on the human leukemia cell line (HL-60) by inducing apoptosis and an increase in chromatin condensation, in association with the decreased expression level of Bcl-2 and increased caspase 3 activation. The expression level of GADD5α and DR5 were also upregulated [206]. The antiangiogenic effect was also displayed by siphonaxanthinin human umbilical vein endothelial (HUVEC) cells and aortic rings of rats [207]. It reduced the mRNA expression level of fibroblast growth factor 2 (FGF-2), fibroblast growth factor receptor (FGFR-1), and early growth response 1 (EGR-1) [208,209].

#### 6.3.5. Zeaxanthin and Lutein

Zeaxanthin (**123**) (Figure 20) is a carotenoid alcohol present in many microalgae, such as *Porphyridium cruentum*, *Isochrysis galbana*, *Phaeodactylum tricornutum*, *Tetraselmis suecica* and *Nannochloropsis gaditana* [210]. It exhibited potent cytotoxicity against human colon adenocarcinoma cell line (HT-29) but it did not induce any cytotoxicity against human normal colon epithelial cell line (CCD 841 CoTr) [211]. Lutein (**124**) is a xanthophyll carotenoid, which exhibited similar anticancer profile as zeaxanthin [211,212].

### 6.4. Stigmasterol

Stigmasterol (**125**) (Figure 20) is a sterol extracted from a microalga benthic diatom *Navicula incerta*. It showed significant anticancer activity by inhibiting the proliferation of the human liver cancer cell line (HepG2) by inducing apoptosis through mitochondrial membrane potential and cause morphological changes and damage of DNA [213,214]. The up-regulation of the expression of caspase-8, caspase-9, Bax, and p53 was induced by stigmasterol whereas antiapoptotic proteins, such as Bcl-2, and X-linked inhibitor of apoptosis protein (XIAP), was down-regulated. The result showed that the cell cycle arrest takes place at G_0_/G_1_ and G_2_/M Phases due to cell component defects.

### 6.5. Nonyl 8-Acetoxy-6-Methyloctanoate

Nonyl 8-acetoxy-6-methyloctanoate (**126**) (Figure 20) is a fatty alcohol ester isolated from a marine diatom *Phaeodactylum tricornutum*. The anticancer activity of the secondary metabolite was established on the human promyelocytic leukemia cell line (HL-60), a human lung carcinoma cell line (A549), and a mouse melanoma cell line (B16F10). It induced damage of DNA and increased the apoptotic activity and triggered cell cycle arrest at the sub G1 Phase. It activated the pro-apoptotic protein Bax, and suppress the antiapoptotic protein Bcl-xL, and also increases the expression levels of both caspase-3 and p53 proteins [215].

### 6.6. Dinochrome A and B

Dinochrome A (**127**) and B (**128**) (Figure 20) are epimeric carotenoids isolated from marine red tide *Peridinium bipes.* They possess strong anticarcinogenic activity by inhibiting the proliferation of GOTO (neuroblastoma cells), OST (osteosarcoma cells) and HeLa cells [216].

### 6.7. Phaeophytins

Phaeophytins are porphyrin-containing organic heterocyclic molecules. Several phaeophytins, such as porphyrinolactone (**129**), 20-chlorinated (13^2^-*S*)-hydroxyphaeophytin A (**130**), (13^2^-*S*)-hydroxyphaeophytin A (**131**) and B (**132**), and (13^2^-*R*)-hydroxyphaeophytin A (**133**) and **B** (**134**) **(**Figure 21) were isolated from a marine green algae *Cladophora fascicularis.* The antiproliferative activity was characterized by inhibition of the activation of NF-κB in the HeLa carcinoma cell line by inhibiting the TNF-α-induced NF-*κ*B translocation from the cytoplasm into the nucleus [217].

### 6.8. Nigricanosides A (135) and B (136) and Methyl Esters of Nigricanosides A (**137**) and B (**138**)

These are glycolipids (Figure 21) extracted from a green algae *Avrainvillea nigricans*. These metabolites inhibited the proliferation of human breast cancer MCF-7 cells and human colon cancer HCT-116 cells and they also possess antimitotic activity which triggers tubulin polymerization within the cells [218].

## 7. Conclusions, Current Challenges and Future Perspectives

This review describes the most recently extracted or generated molecules from marine organisms, such as cyanobacteria and microalgae, with potential for cancer therapy. Marine resources certainly have valuable and undiscovered biochemical versatility and demonstrate a greater opportunity for the development of new anticancer agents. While a variety of compounds have been identified to suppress cell growth in a broad spectrum of cancer cell types, the mechanism of action still remains unclear. A handful of marine molecules have demonstrated possible cytotoxic actions toward specific cancer by inhibition of cell proliferation, its antimitotic behavior (antitubulin impacts), induced apoptosis and inhibition of movement, invasion, or metastatic potential of cancer cells.

While these metabolites have shown potential for cancer treatment, there are several challenges associated with the development of these drugs that need consideration. Oceans certainly provide a large supply of valuable species, but the researchers still cannot access any of these regions. For years, the selection of entities in readily accessible places was preferred. Therefore, the problem is that marine research is not always easily accessible where researchers and the oceanographer need to have strong working ties. Genetic engineering is undergoing development in order to enhance drug production through the conversion of genetic data from the target compound into the host cells. It is an important field for regulation of the isolation and expression of aquatic genes, helping us produce lead compounds from the aquatic ecosystem in a more controlled way. Secondary metabolites are hard to produce independently from cultures, since their development is directly or indirectly dependent on host. Therefore, several of the main genes stay silent while these things are attempted to evolve In vitro. Another big concern is that the development of a specific molecule requires sufficient resources. There is still little research available on the toxicity studies of these marine metabolites in normal cells; this has to be addressed.

Marine cyanobacteria and microalgae tend to be an effective source of anticancer drugs. Nevertheless, more studies are required to understand the basic targets and pathways behind the cytotoxicity of these compounds in cancer cells.

## Figures and Tables

**Figure 1 marinedrugs-18-00476-f001:**
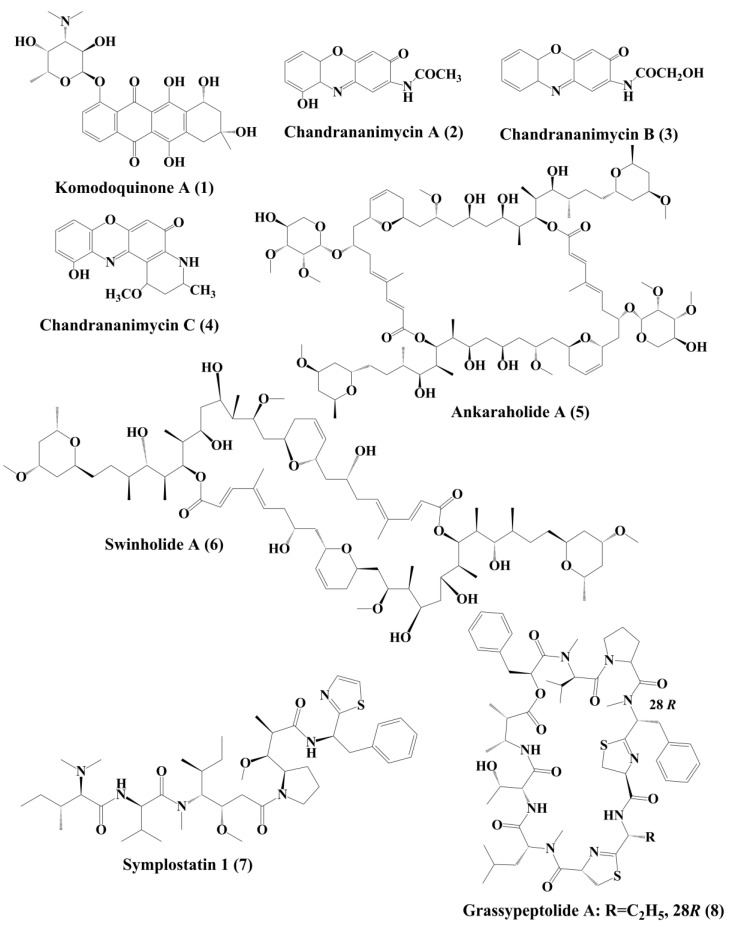
Isolated marine cyanobacteria-derivedanthracyclines, phenoxazin-3-one compounds, polyketides and peptides (**1**–**8**).

**Figure 2 marinedrugs-18-00476-f002:**
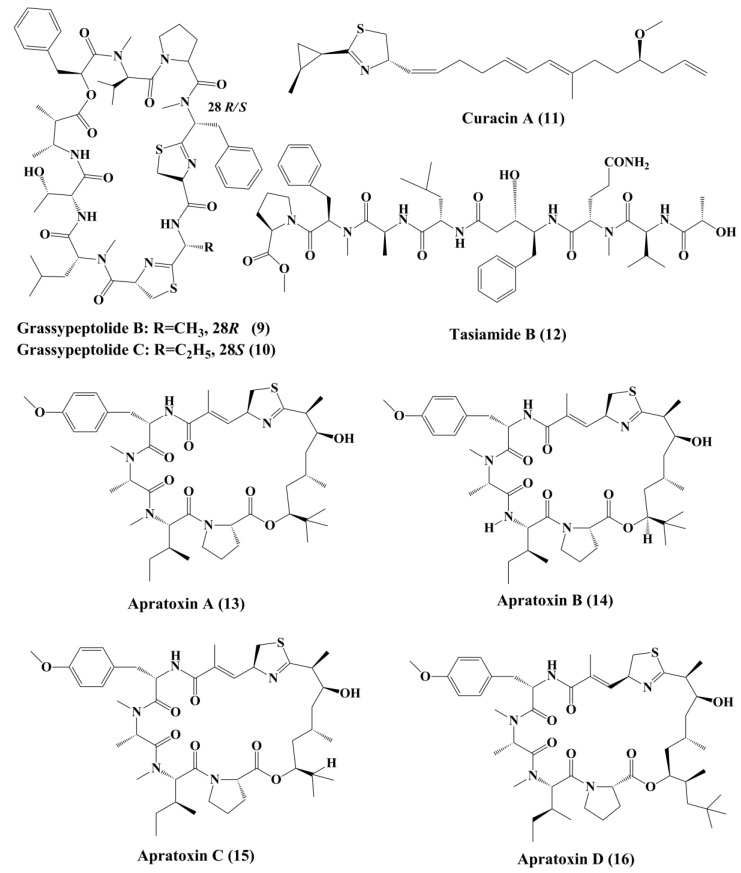
Isolated marine cyanobacteria-derived peptides (**9**–**16**).

**Figure 3 marinedrugs-18-00476-f003:**
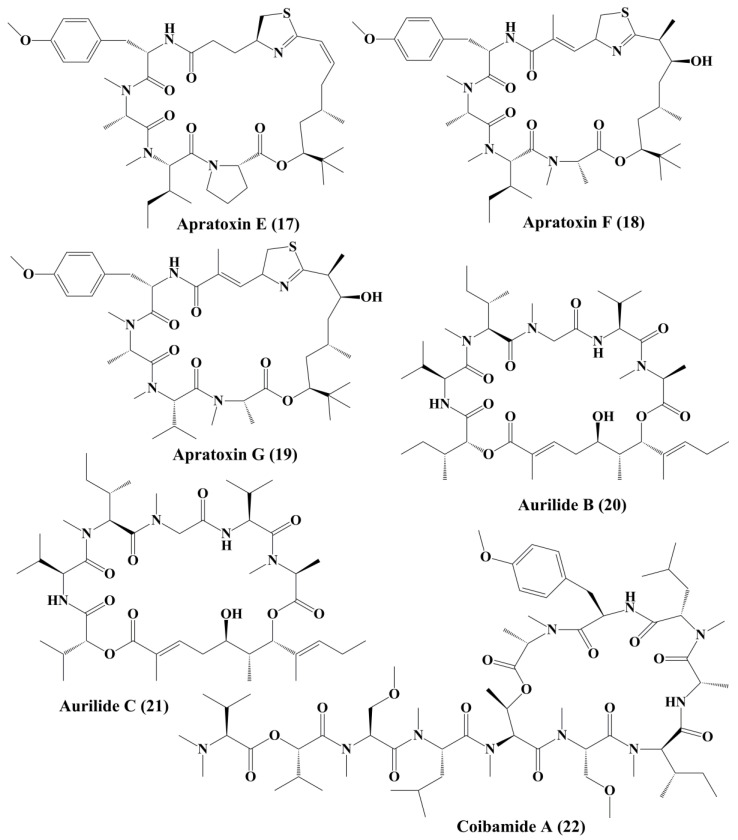
Isolated cyanobacteria-derived peptides (**17**–**22**).

**Figure 4 marinedrugs-18-00476-f004:**
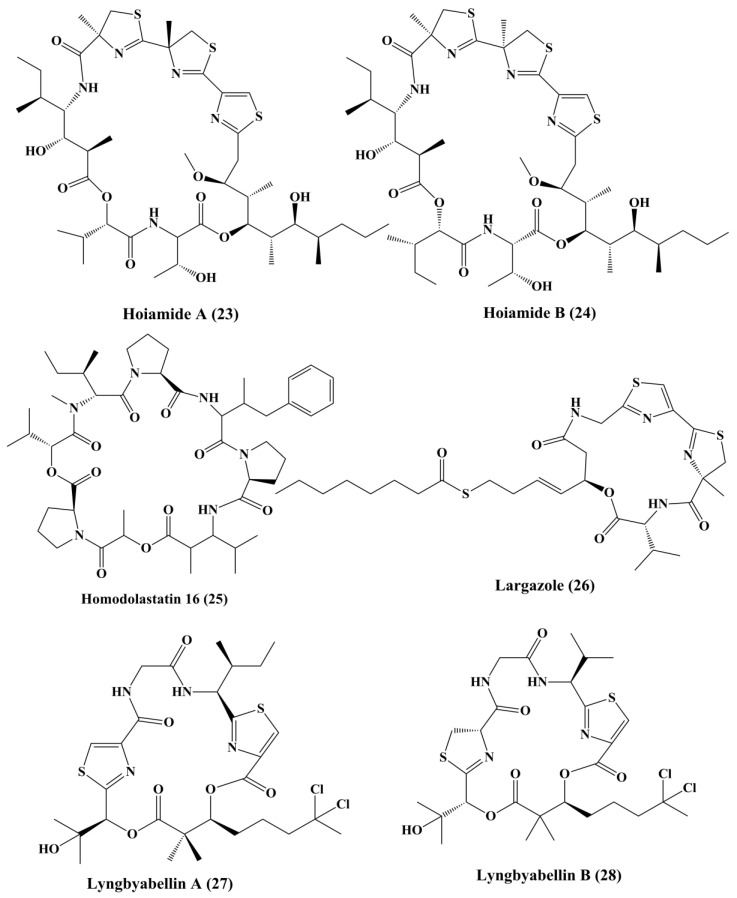
Isolated marine cyanobacteria-derived peptides (**23**–**28**).

**Figure 5 marinedrugs-18-00476-f005:**
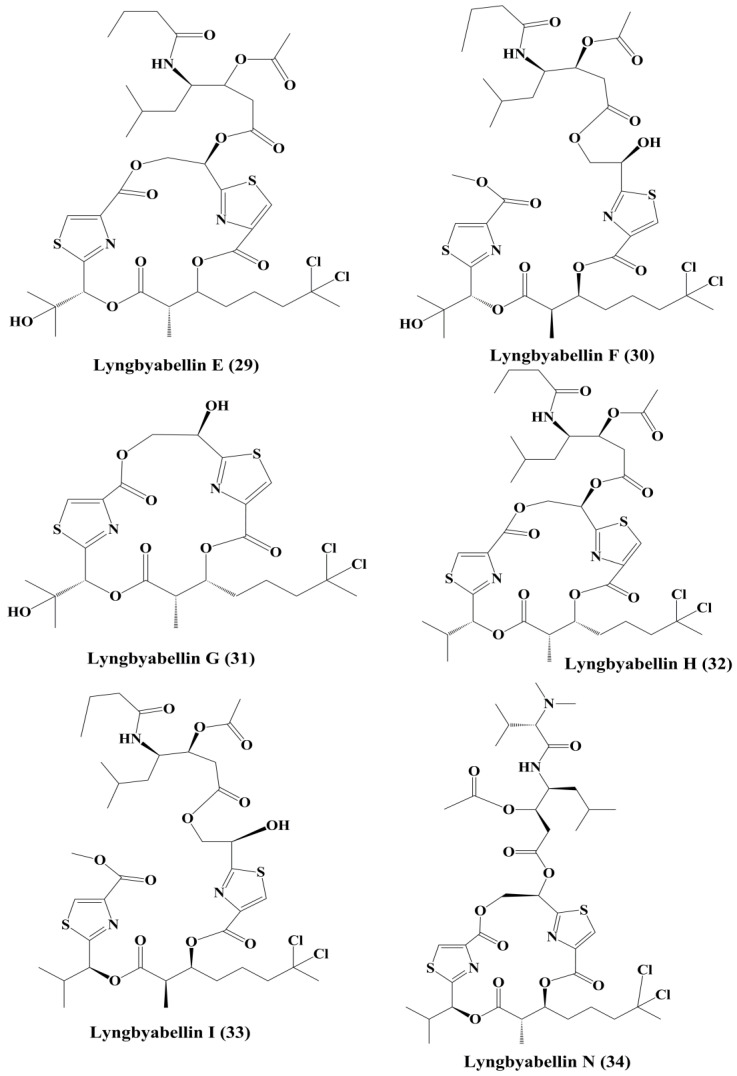
Isolated marine cyanobacteria-derived peptides (**29**–**34**).

**Figure 6 marinedrugs-18-00476-f006:**
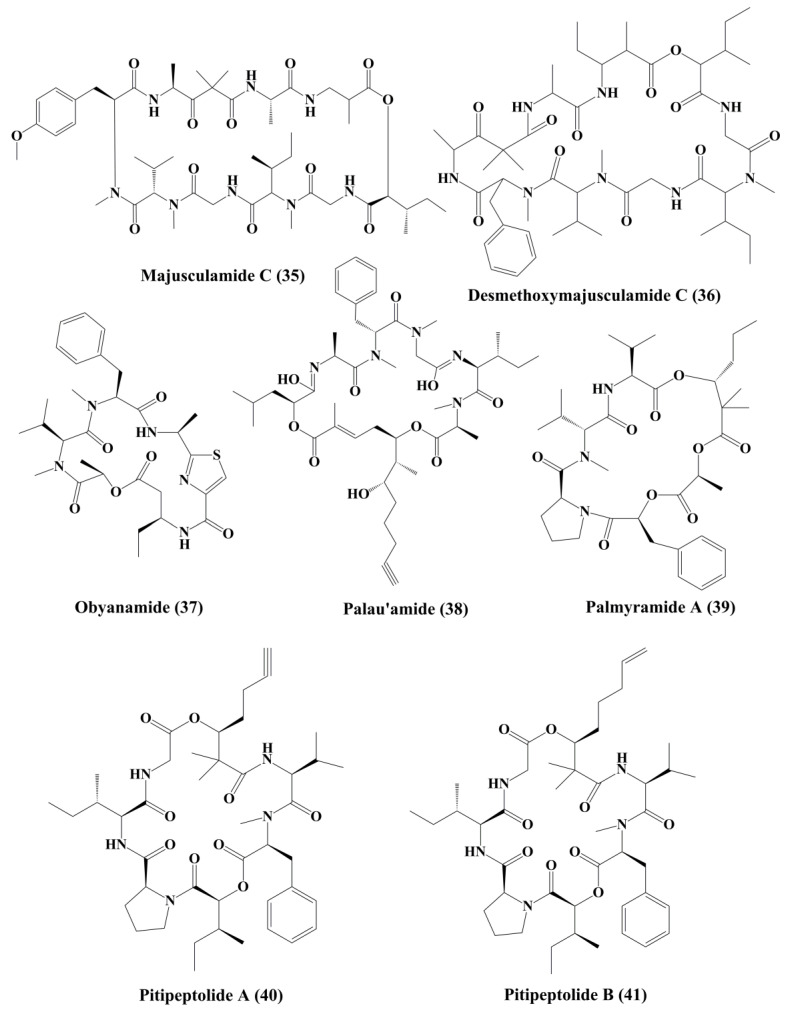
Isolated marine cyanobacteria peptides (**35**–**41**).

**Figure 7 marinedrugs-18-00476-f007:**
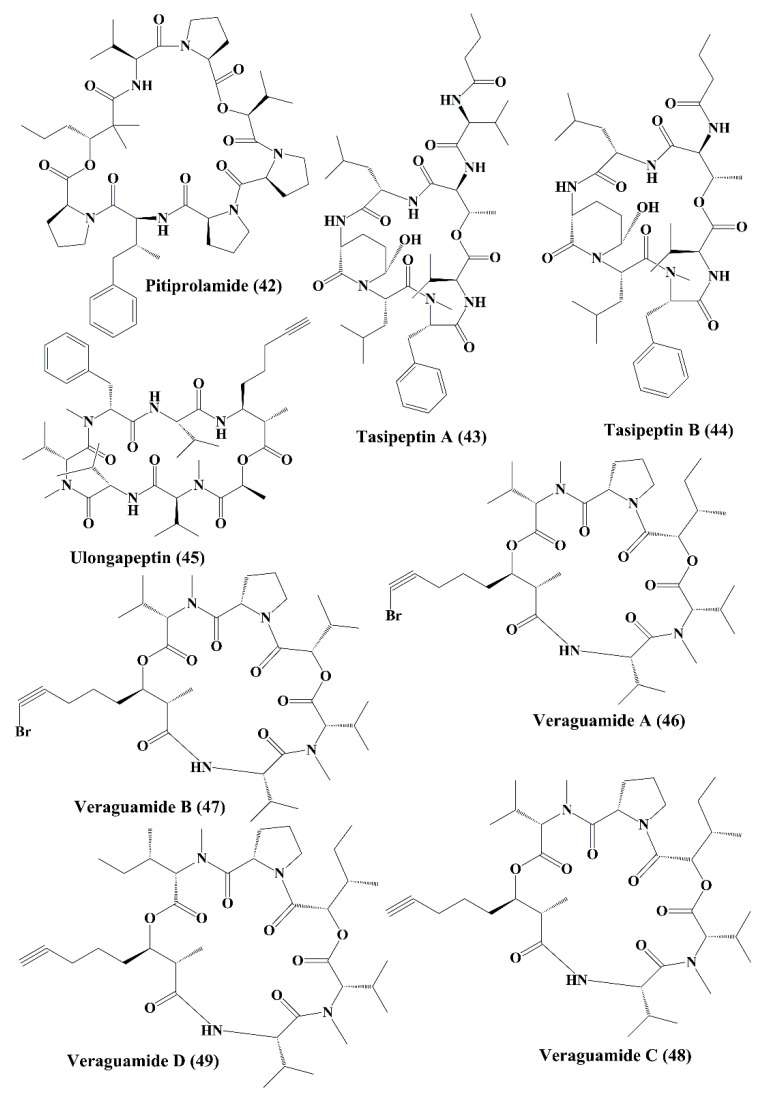
Isolated marine cyanobacteria-derived peptides (**42**–**49**).

**Figure 8 marinedrugs-18-00476-f008:**
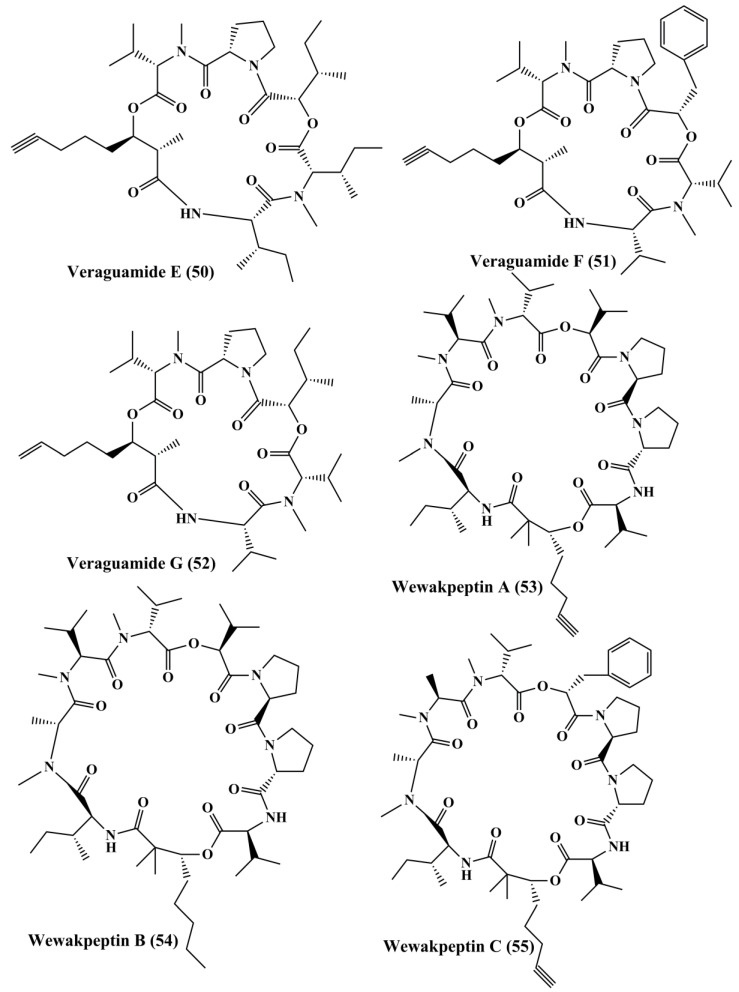
Isolated marine cyanobacteria peptides (**50**–**55**).

**Figure 9 marinedrugs-18-00476-f009:**
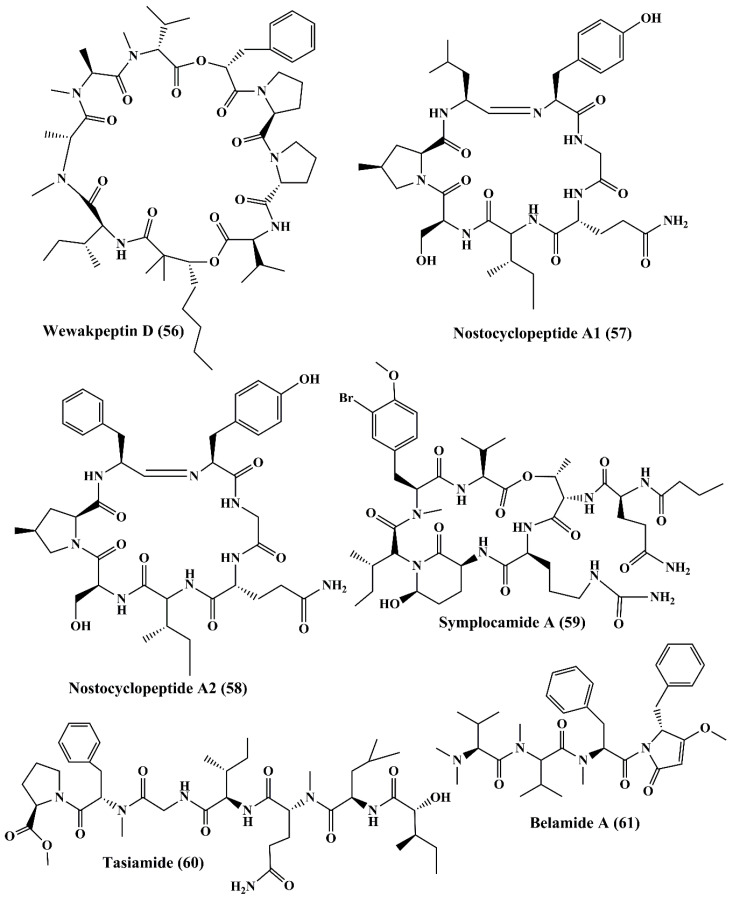
Isolated marine cyanobacteria peptides (**56**–**61**).

**Figure 10 marinedrugs-18-00476-f010:**
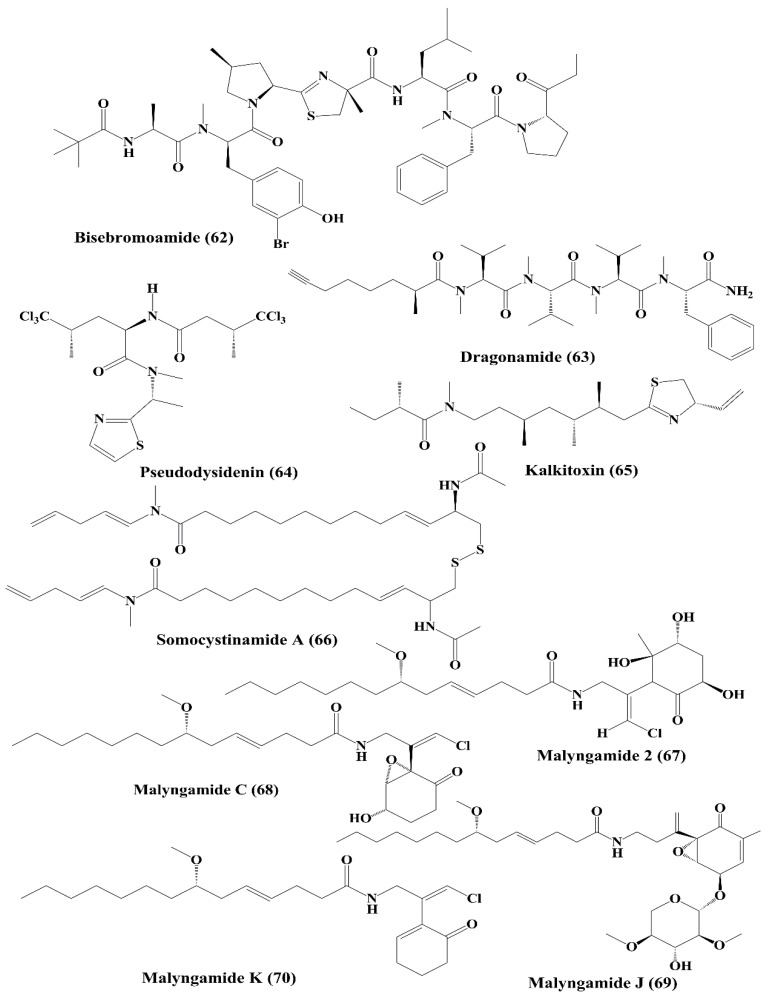
Isolated marine cyanobacteria-derived peptides (**62**–**70**).

**Figure 11 marinedrugs-18-00476-f011:**
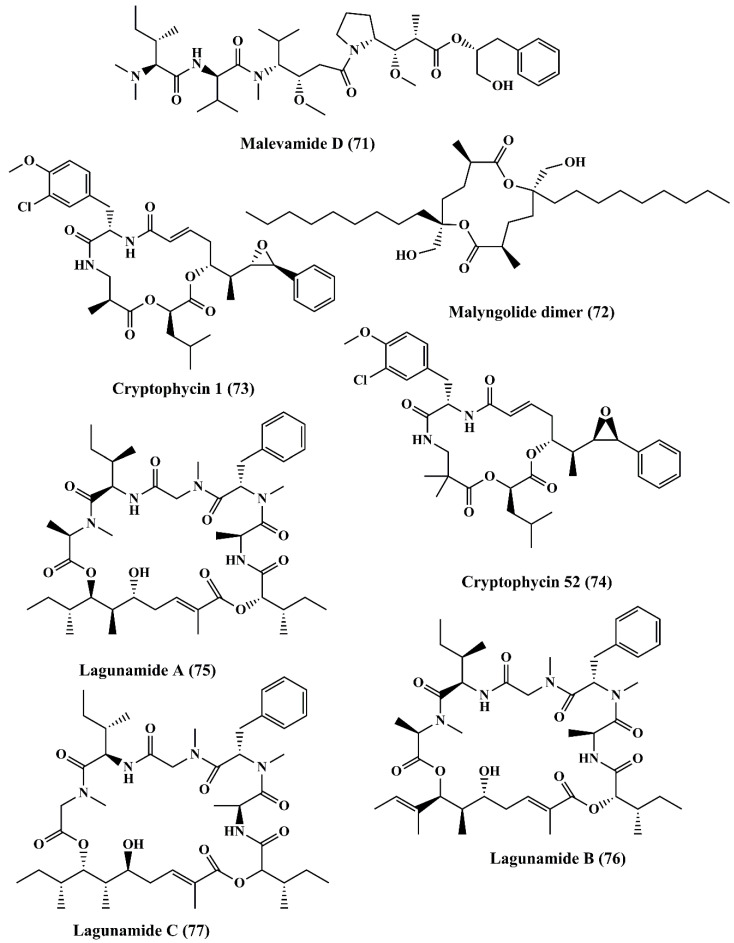
Isolated marine cyanobacteria-derived peptides (**71**–**77**).

**Figure 12 marinedrugs-18-00476-f012:**
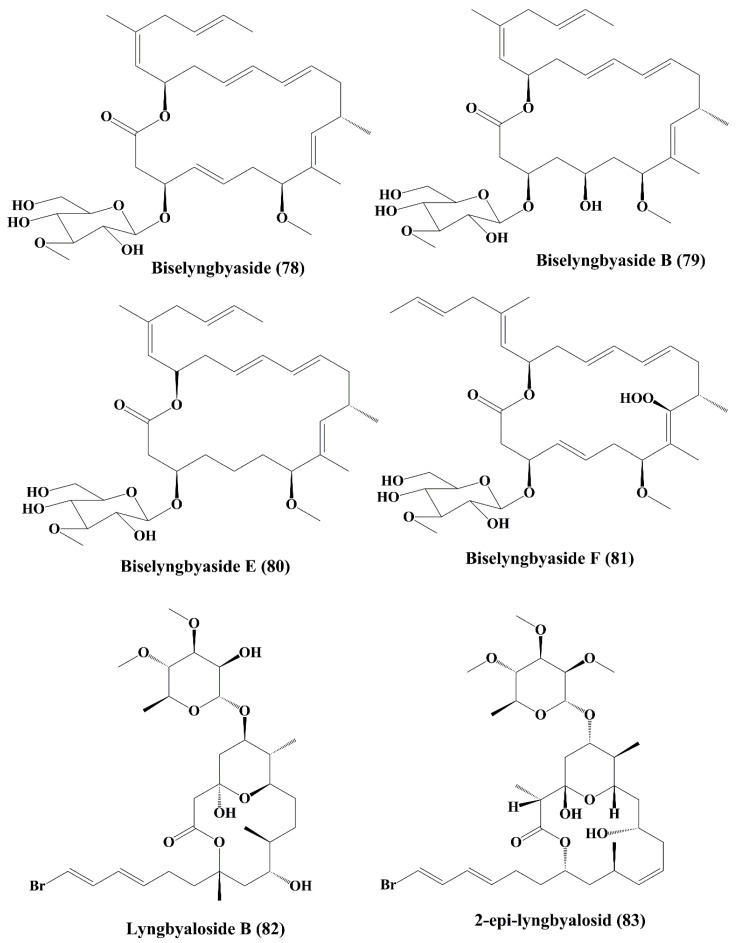
Isolated marine cyanobacteria-derived macrolides (**78**–**83**).

**Figure 13 marinedrugs-18-00476-f013:**
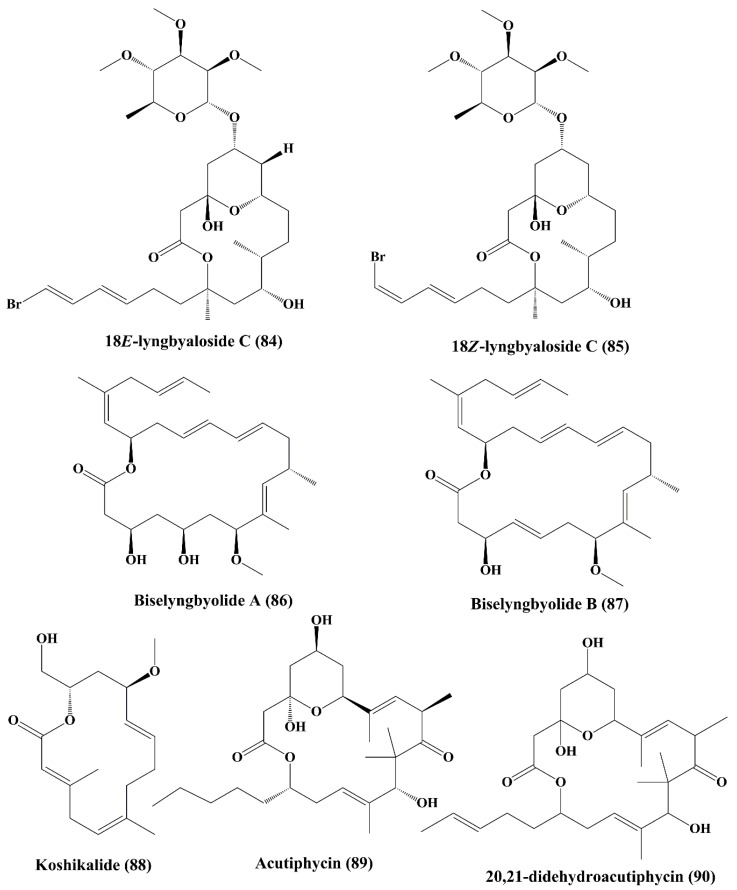
Isolated marine cyanobacteria-derived macrolides (**84**–**90**).

**Figure 14 marinedrugs-18-00476-f014:**
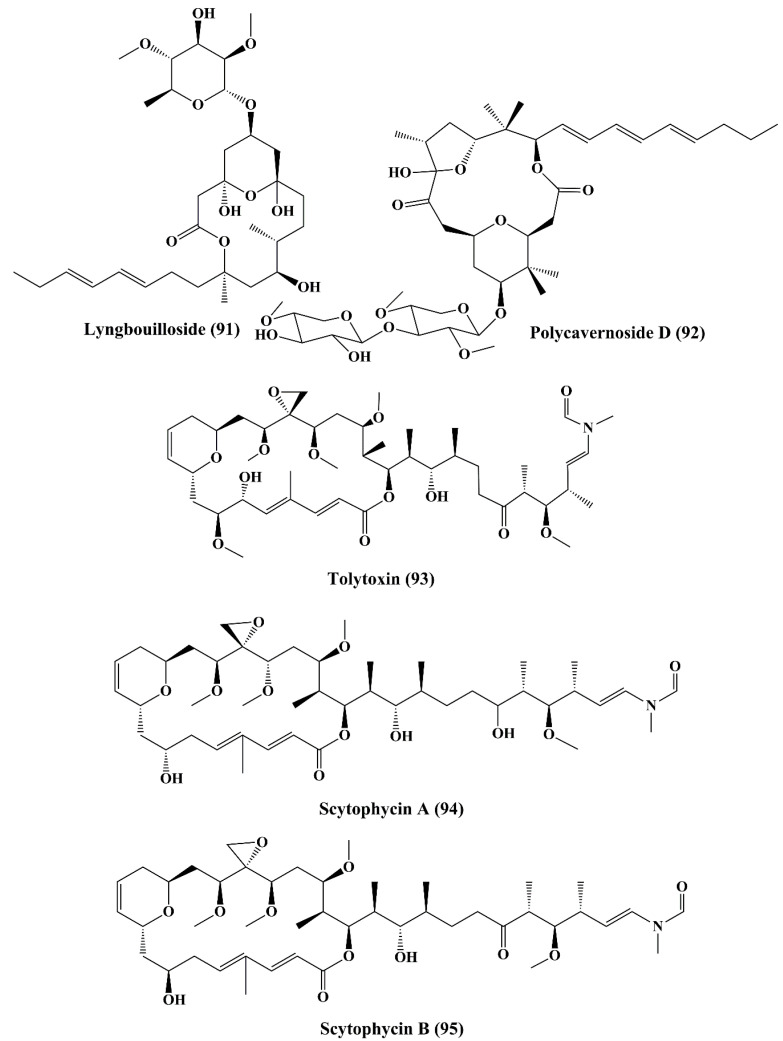
Isolated marine cyanobacteria-derived macrolides and lactones (**91**–**95**).

**Figure 15 marinedrugs-18-00476-f015:**
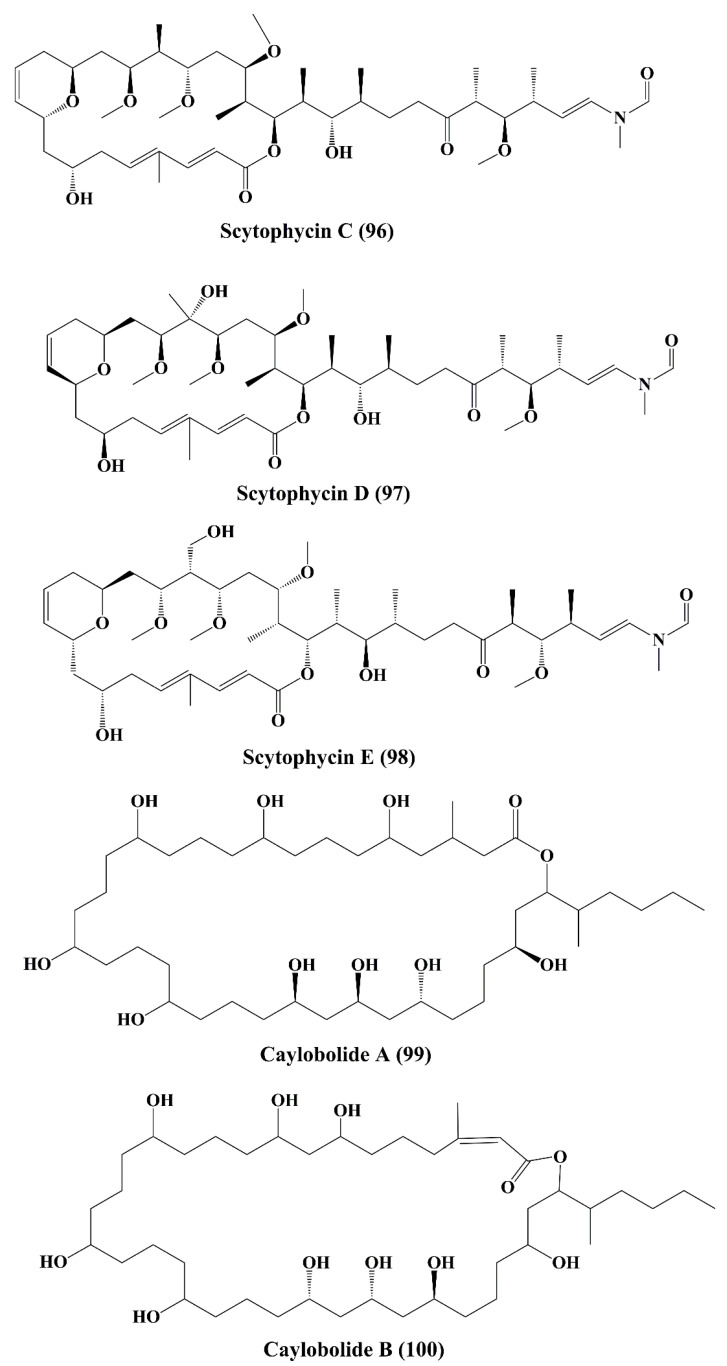
Isolated marine cyanobacteria-derived lactones (**96**–**100**).

**Figure 16 marinedrugs-18-00476-f016:**
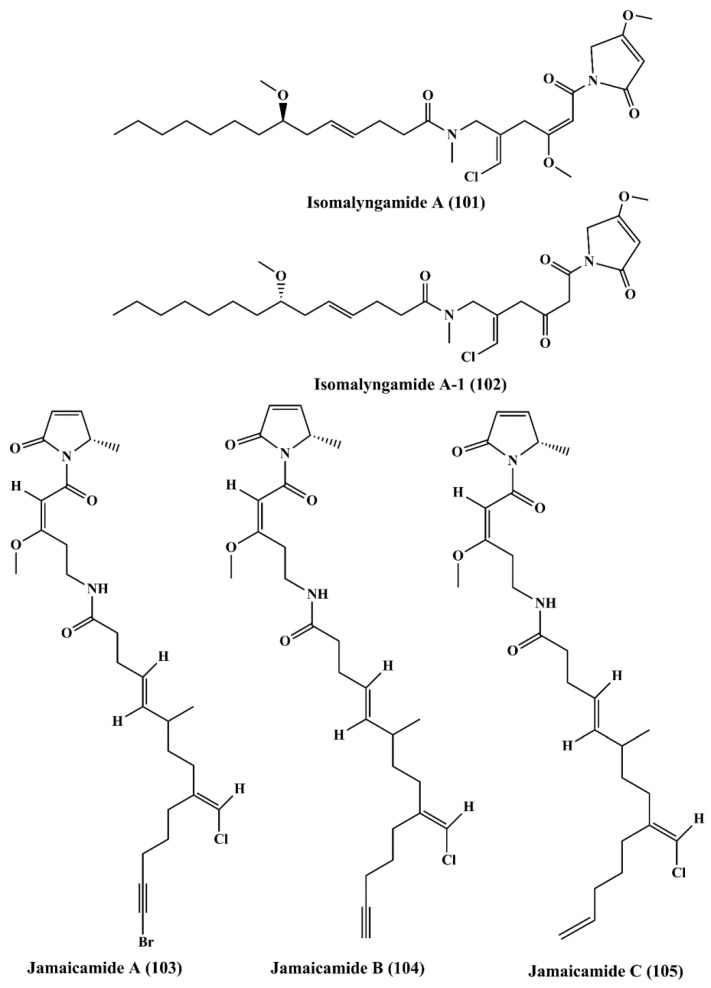
Isolated marine cyanobacteria-derived fatty acid amines (**101**–**105**).

**Figure 17 marinedrugs-18-00476-f017:**
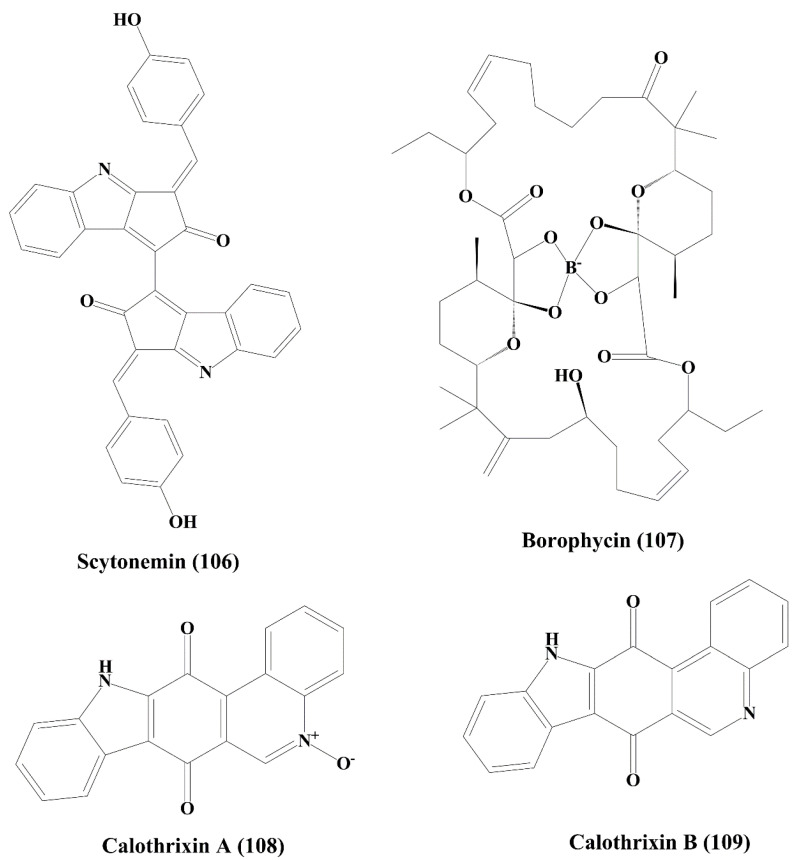
Isolated marine cyanobacteria-derived pigment, boron containing metabolite, and phenanthridine alkaloids (**106**–**109**).

**Figure 18 marinedrugs-18-00476-f018:**
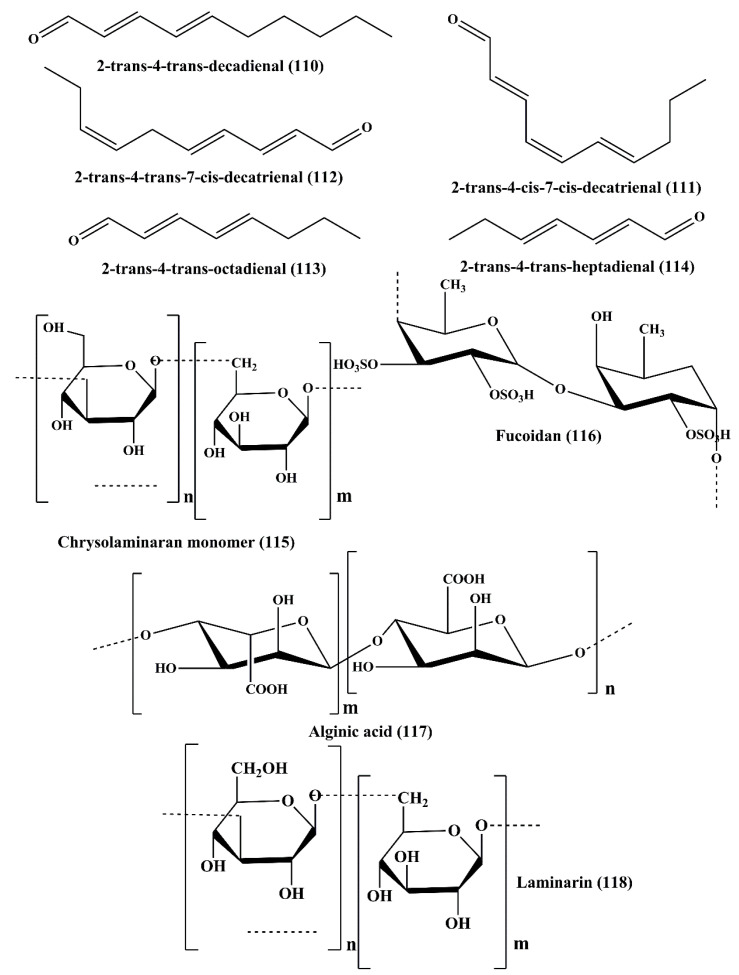
Chemical structure of polyunsaturated aldehydes and polysaccharides (**110**–**118**).

**Figure 19 marinedrugs-18-00476-f019:**
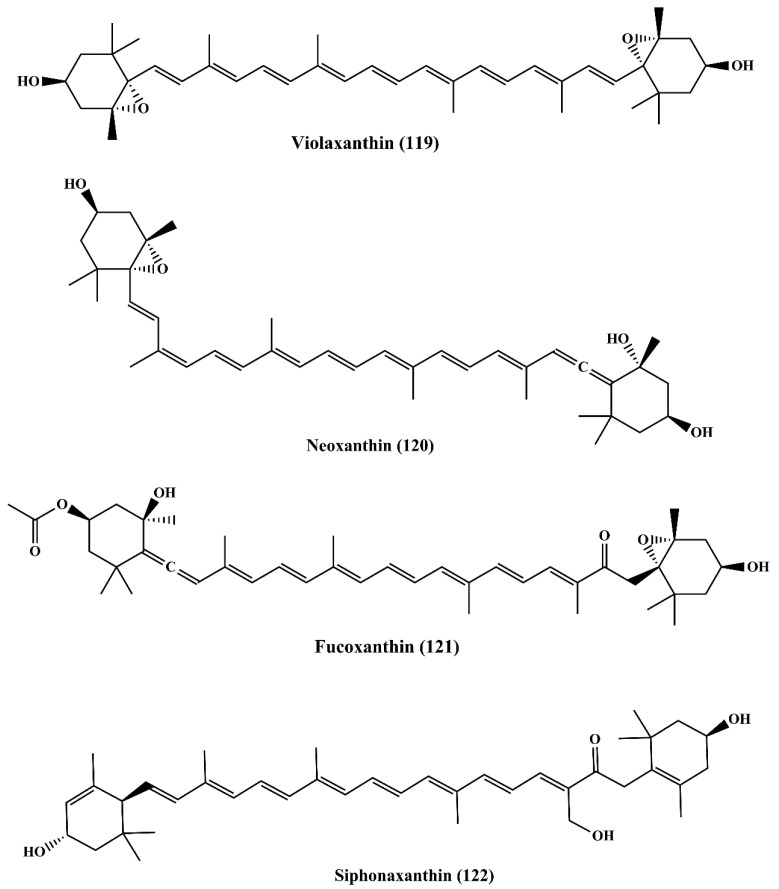
Chemical structures of marine microalgae carotenoids (**119**–**122**).

**Figure 20 marinedrugs-18-00476-f020:**
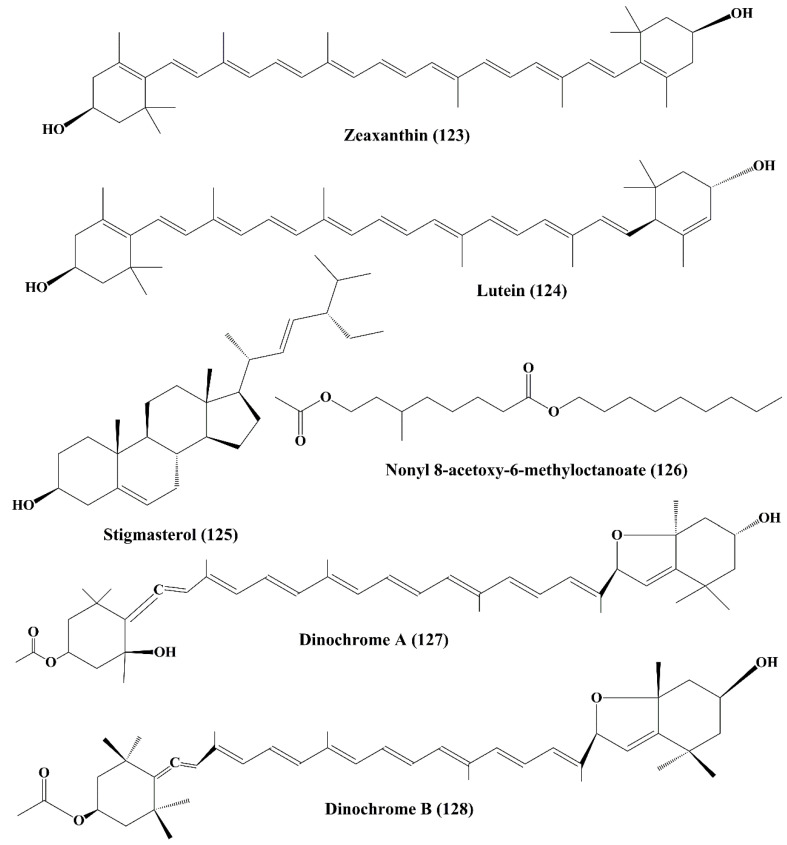
Chemical structure of marine microalgae metabolites (**123**–**128**).

**Figure 21 marinedrugs-18-00476-f021:**
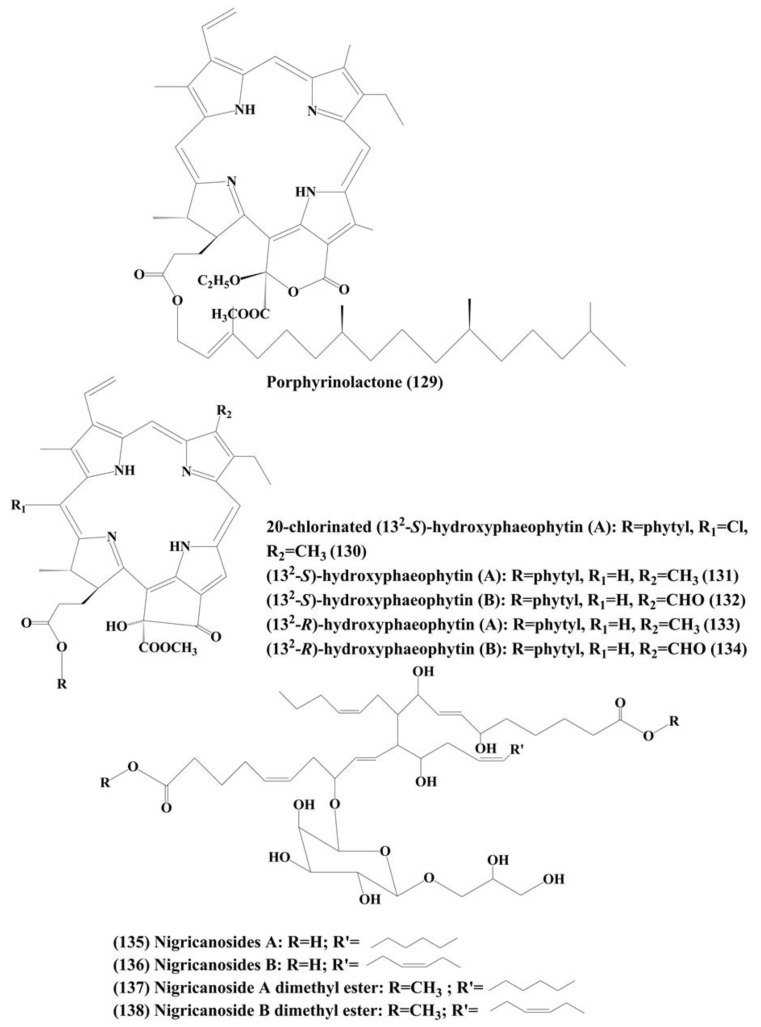
Structure of marine microalgae metabolites (**129**–**138**).

**Table 1 marinedrugs-18-00476-t001:** Anticancer effects and mechanisms of action of various secondary metabolites of marine cyanobacteria.

Class	Secondary Metabolite	Biological Source	Cell Lines Used	Effects and Mechanisms	IC_50_ Values	References
Anthracycline	Komodoquinone A(**1**)	*Streptomyces* sp. KS3	Neuro 2A neuroblastoma cell	Neuritogenic activity, ↑cell differentiation	1 μg/mL	[84]
Phenoxazin-3-one	Chandrananimycins A, B, C (**2**,**3**,**4**)	*Actinomadura* sp.	CCL HT29 (colon cancer); MEXF 514L (melanoma); LXFA 526L, LXFL 529L (lung cancer); CNCL SF268, LCL H460, MACL MCF-7 (breast cancer); PRCL PC3M, RXF 631L (kidney tumor cells)	Anti-tumor activity	~1.4 μg/mL	[85]
Glycosilated polyketide	Ankaraholide A (**5**)	*Geitlerinema* sp.	NCI-H460; Neuro-2a; MDAMB- 435 cell lines	┴ Proliferation; ↑cytotoxicity	119; 262; 8.9 nM	[86]
Polyketide	Swinholide A (**6**)	*Symploca* cf. sp.	Several cancer cell lines	Antitumor activity; ┴ proliferation; ↑cytotoxicity; disruption of the actin cytoskeleton	0.37 nM–1.0 μM	[86]
Pentapeptide	Symplostatin 1 (**7**)	*Symploca hydnoides*	MDA-MB-435 (breast cancer cell), SK-OV-3 (ovarian cancer cell), NCI/ADR (multidrug-resistance ovarian cancer cell), A-10 (smooth muscle cells), and HUVEC (Human umbilical vein endothelial cells); in vivo study (murine colon 38 and murine mammary 16/C carcinoma cells)	Antitumor activity; ↑phosphorylation of Bcl-2; ↑micronuclei formation, ↑caspase 3, ↑apoptosis, cell cycle arrest at G2/M Phase, ┴tubulin accumulation	0.15 ± 0.03 nM; 0.09 ± 0.02 nM; 2.90 ± 0.64 nM; 1.8 ± 0.43 nM; 0.16 ± 0.02 nM	[72,87]
Macrocyclic depsipeptide	Grassypeptolide, Grassypeptolide A, B and C (**8**, **9**, **10**)	*Lyngbya confervoides*	human osteosarcoma (U2OS), cervical carcinoma (HeLa), colorectal adenocarcinoma (HT29), and neuroblastoma (IMR-32);	Anticancer activity; ┴proliferation; Cell cycle arrest at G1 or G2/M Phase	1–4.2 μM for grassypeptolide in all cell lines. Grassypeptolide A: 1.22 &1.01 μM in HT29 and Hela. Grassypeptolide B: 4.07 and 2.93 μM in HT29 and Hela. Grassypeptolide C: 76.7 and 44.6 nM HT29 and Hela.	[88,89]
ketopeptide	Curacin A (**11**)	*Lyngbya majuscule*	Non-small cell lung cancer cell line (A549)	Anticancer activity; ┴proliferation; ↑apoptosis; cell cycle arrest at G2/M Phase; binds to tubulin at colchicines binding site	0.72 ± 0.02 μM	[90,91,92,93,94,95]
Linear peptide	Tasiamide B (**12**)	*Symploca* sp.	ĸB oral epidermoid cancer; human colon carcinoma (LoVo) cells	┴Proliferation; ↑cytotoxicity	0.48; 3.47 μg/mL	[96]
Cyclic depsipeptide	Apratoxin A (**13**)	*Lyngbya majuscula*	U2OS osteosarcoma; HeLa cervical carcinoma; in LoVo colon carcinoma; ĸB carcinoma cancer cells	┴Secretory pathway; ┴cell cycle at G1 Phase; ↑cytotoxicity; ┴translocation of protein targeting Sec61α	50; 2.2; 0.36; 0.52 nM	[97,98]
	Apratoxin B (**14**)	*Lyngbya* sp.	ĸB oral epidermoid cancer and LoVo colon cancer lines	↑Cytotoxicity	21.3; 10.8 nM	[97]
	Apratoxin C (**15**)	*Symploca* cf. sp.	Several cancer cell lines	↑Cytotoxicity	1.0; 0.73 nM	[97]
	Apratoxin D (**16**)	*Lyngbya majuscule*; *Lyngbya sordida*	H-460 lung cancer		2.6 nM	[98]
	Apratoxin E (**17**)	*Lyngbya bouilloni*	U2OS osteosarcoma, HT29 colon adenocarcinoma and HeLa epithelial carcinoma	↑Antiproliferative Activity	59; 21; 72 nM	[99]
	Apratoxin F (**18**)	*Lyngbya* sp.	H-460 lung cancer; HCT-116 colorectal cancer cells	↑Cytotoxicity	2; 36.7 nM	[100]
	Apratoxin G (**19**)				14 nM; Not specified	
	Aurilide B (**20**)	*Lyngbya majuscula*	NCI-H460 human lung tumor and neuro-2*a* mouse neuroblastoma cells	↑Antiproliferative activity; ↑OPA1 synthesis, ↑apoptosis	0.04; 0.01 µM	[101,102]
	Aurilide C (**21**)	*Lyngbya majuscula*	U2OS osteosarcoma, HT29 colon adenocarcinoma and HeLa epithelial carcinoma		0.13; 0.05 µM	
	Coibamide A (**22**)	*Leptolyngbya* sp.	MDA-MB-231, melanoma LOX IMVI, leukemia HL-60 and astrocytoma SNB75	↑Cytotoxicity; ┴cell cycle at G1 Phase	2.8; 7.4; 7.4 and 7.6 nM	[103]
			glioblastoma cell lines U87-MG and SF-295	↑Cytotoxicity	20 nM	[104]
			Normal human umbilical vein endothelial cells (HUVECs)	┴Proliferation; ↓VEGFR2	0.3–3 nM	
			Human U87-MG glioblastoma cells and SF-295 glioblastoma cells	↑Cytotoxicity; ↑autophagy	28.8, 96.2 nM	[105]
	Hoiamide A (**23**)	*Lyngbya majuscule*, *Phormidium gracile*	H-460 lung cancer and neuro-2a mouse neuroblastoma	↑Cytotoxicity; ↑neurotoxicity	11.2; 2.1 μM	[106]
	Hoiamide B (**24**)				8.3 μM; no effect on neuro-2a	
	Homodolastatin 16 (**25**)	*Lyngbya majuscule*	WHCO1 and WHCO6 esophageal cancer; ME180 cervical cancer	↑Apoptosis; ┴cell cycle at G2/M Phase; ↑cytotoxicity	4.3 and 10.1; 8.3 µg/mL	[107]
	Largazole (**26**)	*Symploca* sp.	MDA-MB-23I breast cancer; U2OS osteosarcoma; colon HT29; neuroblastoma IMR-32; nontransformed murine mammary epithelial cells NMuMG; HCT-116 colorectal carcinoma	↑Cytotoxicity; ┴tumor; cell cycle arrest at G2/M Phase; ┴HDAC	7.7; 55; 12; 16; 122 nM; Not specified	[108,109]
	Lyngbyabellin A (**27**)	*Lyngbya majuscula*	ĸB nasopharyngeal carcinoma and LoVo colon adenocarcinoma	↑Cytotoxicity; ┴tumor; cell cycle arrest at G2/M Phase; ↑actin polymerization	0.03; 0.05 μg/mL	[110]
	Lyngbyabellin B (**28**)	*Lyngbya majuscula*			0.10; 0.83 μg/mL	[110]
	Lyngbyabellin E (**29**)	*Lyngbya majuscula* *Symploca* sp.	NCI-H460 human lung tumor and neuro-2a mouse neuroblastoma cells	┴Tumor growth; ┴cell microfibrils network	0.4; 1.2 μM	[111]
	Lyngbyabellin F (**30**)	*Lyngbya majuscula*		↑Cytotoxicity	1; 1.8 μM	
	Lyngbyabellin G (**31**)	*Lyngbya majuscula*			2.2; 4.8 μM	[111]
	Lyngbyabellin H (**32**)				0.2; 1.4 μM	[111]
	Lyngbyabellin I (**33**)	*Lyngbya majuscula*			1; 0.7 μM	[111]
	Lyngbyabellin N (**34**)	*Moorea bouillonii*	HCT116 (colon cancer cell line)	Anticancer activity; ↑cytotoxicity	40.9 ± 3.3 nM	[112]
	Majusculamide C (**35**)	*Lyngbya majuscule*	Ovarian carcinoma OVCAR-3, kidney cancer A498, lung cancer NCI-H460, colorectal cancer KM20L2; glioblastoma SF-295	Anticancer activity; ↑cytotoxicity	0.51; 0.058; 0.0032; 0.0013; 0.013 μg/mL	[110,113]
	Desmethoxymajusculamide C (**36**)	*Lyngbya majuscule*	HCT-116 human colon carcinoma cells	Selective antitumor activity	20 nM	[110]
	Obyanamide (**37**)	*Lyngbya confervoides*	ĸB and LoVo cells	Anticancer activity	0.58; 3.14 µg/mL	[114]
	Palau’amide (**38**)	*Lyngbya confervoides*	ĸB oral epidermoid cancer cells	Anticancer activity	13 nM	[115]
	Palmyramide A (**39**)	*Lyngbya majuscule*	Neuro2a cells and human lung cell H-460	Anticancer activity; ↑cytotoxicity; blocking the voltage regulated sodium channel	17.2; 39.7 µM	[116]
	Pitipeptolide A (**40**)	*Lyngbya majuscule*	HT29 colon adenocarcinoma cancer cells, MCF-7 and LoVo colon cancer	Anticancer activity; ↑cytotoxicity	13; 13 µM & 2.25 µg/mL	[117,118]
	Pitipeptolide B(**41**)	*Lyngbya majuscula*	HT29 colon adenocarcinoma cancer cells, MCF-7 and LoVo colon cancer	Anticancer activity; ↑cytotoxicity	13; 11 µM; 1.95 µg/mL	[117,118]
	Pitiprolamide (**42**)	*Lyngbya majuscula*	HCT116 colorectal carcinoma and MCF7 breast adenocarcinoma	Anticancer activity, ↑cytotoxicity	33; 33 µM	[119]
	Tasipeptins A (**43**)	*Symploca* sp.	ĸB oral epidermoid cancer	Anticancer activity, ↑cytotoxicity	0.93 µM	[120]
	Tasipeptins B (**44**)	*Symploca* sp.	ĸB oral epidermoid	Anticancer activity, ↑cytotoxicity	0.82 µM	[120]
	Ulongapeptin (**45**)	*Lyngbya* sp.	ĸB oral epidermoid cancer	Anticancer activity; ↑cytotoxicity	0.63 µM	[121]
	Veraguamide A-G (**46**–**52**)	*Symploca* cf. *hydnoides*, *Oscillatoria margaritifera*	HT29 colon adenocarcinoma; HeLa cervical carcinoma	Anticancer activity; ↑cytotoxicity	26; 2 µM & 141 nM; 30 & 17 µM; 5.8 & 6.1 µM; 0.84 & 0.54 µM; 1.5 & 0.83 µM; 49 & 49 µM; 2.7 & 2.3 µM	[122,123]
	Wewakpeptins A-D (**53**–**56**)	*Lyngbya semiplena*	H-460 lung cancer	Anticancer activity; ↑cytotoxicity	0.4 µM	[124]
Cyclic heptapeptides	Nostocyclopeptide A1 & A2 (**57**, **58**)	*Nostoc* sp.	ĸB oral epidermoid cancer and LoVo colon carcinoma cell line	Anticancer activity; ↑cytotoxicity	1 & 1 µM for both	[125]
Cyclopeptide	Symplocamide (**59**)	*Symploca* sp.	Non-small cell lung cancer cells H-460 and neuro-2a neuroblastoma cells	Anticancer activity; ↑cytotoxicity	40; 29 nM	[110]
Cyclicpeptide	Tasiamide (**60**)	*Symploca* sp.	Human nasopharyngeal carcinoma (ĸB) and human colon carcinoma (LoVo) cells	Anticancer activity; ↑cytotoxicity	0.48; 3.47 µg/mL	[126]
Linear tetrapeptide	Belamide A (**61**)	*Symploca* sp.	MCF7 breast cancer cell; HCT-116 colon cancer cell	Anticancer activity; ↑cytotoxicity; depolymerizing effect on microtubule in A-10 cells; antimitotic activity	1.6 µM; 0.74 µM	[127]
Peptide	Bisebromoamide (**62**)	*Lyngbya* sp.	HeLa S3 cells; a panel of 39 human cancer cell lines of the Japanese Foundation for Cancer Research (JFCR39) Cancer Research	↑Cytotoxicity; ┴protein kinases; ┴phosphorylation of ERK	0.04 µg/mL; average 40 nM	[128,129]
Lipopeptides	Dragonamide, Pseudodysidenin (**63**, **64**)	*Lyngbya majuscula*	P-388; A-549 lung epithelial adenocarcinoma, HT-29 colon adenocarcinoma; MEL-28 melanoma	Anticancer activity; ↑cytotoxicity	> 1 µg/mL	[130]
Lipopeptide	Kalkitoxin (**65**)	*Phormidium* sp.	HCT-116 colon cancer cell; T47D breast tumor cells	Anticancer activity; ↑cytotoxicity; ┴hypoxia-induced activation of HIF-1; ↓mitochondrial oxygen consumption at electron transport chain (ETC) complex I (NADH-ubiquinone oxidoreductase); blocking of VEGF	2.7 nM; 5.6 nM	[131]
Lipopeptide	Somocystinamide A (**66**)	*Lyngbya majuscula*	Jurkat, CEM (leukemia), A549 (lung carcinoma), Molt4 (T cell leukemia), M21 melanoma, and U266 myeloma cell lines	↑Cytotoxicity; ↑apoptosis via caspase 8	3; 14; 46; 60 nM; 1.3; 5.8 µM	[132]
Lipopeptide, Lyngbic acid derivative	Malyngamide 2 (**67**)	*Lyngbya sordida*	H-460 lung cancer	↑Cytotoxicity	27.3 µM	[133]
	Malyngamide C, J, & K (**68**, **69**, **70**)	*Lyngbya majuscula*	NCI-H460, Neuro-2a, and HCT-116	↑Cytotoxicity	1.4; 3.1; 0.2 µg/mL 10.8, 4 µg/mL, nd 1.1; 0.49 µg/mL, nd	[134]
Peptide ester	Malevamide D (**71**)	*Symploca hydnoides* Kü tzing ex Gomont	P388, Lung cancer A-549, colon cancer HT-29 Melanoma MEL-28	↑Cytotoxicity	0.3–0.7 nM 0.7 nM	[135]
Cyclodepside	Malyngolide dimer (**72**)	*Lyngbya majuscule*	NCI H-460 human lung tumor cell line	Moderate cytotoxicity; anticancer activity	Not specified	[136]
Macrolide depsipeptide	Cryptophycin 1 (**73**)	*Nostoc* sp.	L1210 murine leukemia cells	Anticancer activity; ↑disruption of microtubule assembly	Not specified	[137,138]
			kB cells and LoVo cell	↑Apoptosis	4.58, 7.63 pM	[139]
			MDA-MB-435 mammary adenocarcinoma; SKOV3 ovarian carcinoma cell lines	┴Proliferation; ┴cell cycle at G2/M Phase	50 pM	[140,141]
Cyclic depsipeptide	Lagunamides A, B (**75**, **76**)	*Lyngbya majuscule*	P388 (a murine leukemia cell line)	↑Cytotoxicity	6.4 and 20.5 nM	[142]
	Lagunamides C (**77**)		P388, A549, PC3, HCT8, and SK-OV3carcinoma cell lines		2.1 to 24.4 nM	[143]
Macrolide glycoside	Biselyngbyaside (**78**)	*Lyngbya* sp.	HeLa S_3_ epithelial carcinoma; SNB-78 central nervous system cancer; NCI H522 lung cancer	┴Proliferation of cancer cell; induced cytotoxicity	0.1 µg/mL; 0.036; 0.067 µM	[144]
	Biselyngbyasid B (**79**)	*Symploca hydnoides*	HeLa S_3_ cells and HL60 cells	┴Proliferation of cancer cell; induced cytotoxicity	3.5 & 0.82 µM	[145]
	Biselyngbyasid E & F (**80**, **81**)	*Lyngbya* sp.	HeLa and HL60 cells	┴Proliferation of cancer cell; induced cytotoxicity	0.19 & 0.071 µM; 3.1 & 0.66 µM	[146]
Glycomacrolide	Lyngbyaloside B (**82**)	*Lyngbya* sp.	ĸB nasopharyngeal carcinoma and LoVo colon adenocarcinoma	↑Cytotoxicity; anticancer activity	4.3; 15 µM	[147]
	2-epi-lyngbyalosid (**83**)	*Lyngbya bouillonii*	HT29 colorectal adenocarcinoma and HeLa cells	Anticancer activity; ┴proliferation	38 and 33 µM	[148]
	18E-lyngbyaloside C; 18Z-lyngbyaloside C (**84**, **85**)	*Lyngbya* sp.	HT29 colorectal adenocarcinoma and HeLa cells	Anticancer activity; ┴proliferation;	13 & 9.3 µM; >100 µM & 53 µM	[148]
Macrolide	Biselyngbyolide A; Biselyngbyolide B (**86**, **87**)	*Lyngbya* sp.	HeLa S_3_ cells and HL60 cells	Anticancer activity	0.22 & 0.027 µM; 0.028 & 0.0027 µM	[149]
Macrolide	Koshikalide; Acutiphycin and 20, 21-didehydroacutiphycin (**88**, **89**, **90**)	*Lyngbya* sp., *Oscillatoria acutissima*	HeLa S3 cells; KB and NIH/3T3 cells	Anticancer activity; ↑cytotoxicity	42 µg/mL, Not specified for Acutiphycin and 20, 21-didehydroacutiphycin	[150,151]
Glycosylated macrolide	Lyngbouilloside (**91**)	*Lyngbya bouillonii*	Neuro-2a neuroblastoma cells	Anticancer activity; ↑cytotoxicity	17 µM	[152]
Glycosylated macrolide	Polycavernoside D (**92**)	*Okeania* sp.	H-460 human lung cancer cell line	┴Proliferation	EC_50_ = 2.5 µM	[153]
Macrocyclic lactone	Tolytoxin(**93**) 6-hydroxyscytophycin B (**95**), 19-*O*-demethylscytophycin C (**96**), and 6-hydroxy-7-*O*-methylscytophycin E (**98**)	*Seytonema ocellaturn* Lyngbye ex Bornet and Flahault	L1210 (murine leukemia), LoVo, kB, HEp-2 (human epithelial type 2 cells), HL-60 (Human promyelocytic leukemia), HBL-100 (breast cancer cell), T47-D (human ductal carcinoma), COLO-201 (colon adenocarcinom), KATO-III (human gastric carcinoma) Nasopharynx cell (ĸB cells), &LoVo cells	Anticancer activity; ↑cytotoxicity;	3.9, 8.4, 5.3, 2.3, 4.8, 2.4, 4.9, 0.52, and 0.78 nM >5 ng/mL	[154,155]
Macrolactone	Caylobolide A (**99**), Caylobolide B (**100**)	*Lyngbya majuscula* *Phormidium* sp.	HCT-116 colon tumor HT29 colorectal adenocarcinoma, and HeLa cervical carcinoma	Anticancer activity; ↑cytotoxicity	9.9 µM (same for both caylobolide A & B) 4.5; 12.2 µM	[156,157]
Fatty acid amines	Isomalyngamide A (**101**), and Isomalyngamide A-1 (**102**)	*Lyngbya majuscula*	Breast cancer MCF-7 and MDA-MB-231	┴Proliferation; ┴apoptosis; ┴cell migration; antimetastatic activity	4.6 & 2.8 µM; 12.7 µM & > 20 µM	[158]
	Jamaicamides A, B, & C (**103**, **104**, **105**)	*Lyngbya majuscula*	H-460 lung cancer and Neuro-2a mouse neuro blastoma cell lines	┴proliferation	LC_50_: 15 µM for all	[159]
Pigment	Scytonemin (**106**)	*Stigonema* sp.	Jurkat T cells	↑Apoptosis; ┴formation of mitotic spindle; ┴protein serine/threonine kinase activity	7.8 μM	[160,161]
Boron containing metabolite	Borophycin (**107**)	*Nostoc spongiaeforme*, *N. linckia*	Human cancer cell lines ĸB colorectal adenocarcinoma and LoVo (human epidermoid carcinoma)	┴Cancer; ┴cell cycle at G2/M Phase	Not specified	[48,162]
Phenanthridine alkaloids	Calothrixins A and B (**108**, **109**)	*Calothrix* sp.	Human carcinoma cell line (HeLa)	↑Cytotoxicity;┴proliferation	40and 350 nM	[163,164]
			CEM leukemia cells	┴Proliferation;┴cell cycle at G1 and G2/M Phases	0.20 to 5.13 µM	[165]

Various symbols (↑, ↓ and ┴) indicate increase, decrease and inhibition in the obtained variables, respectively.

**Table 2 marinedrugs-18-00476-t002:** Anticancer effects and mechanisms of action of various secondary metabolites of marine microalgae.

Class	Secondary Metabolite	Biological Source	Cell Lines	Effects and Mechanisms	IC_50_/Conc.	References
Polyunsaturated aldehydes	2-trans-4-trans-decadienal (**110**)	*Thalassiosira rotula*, *Skeletonema costatum*, *Phaeocystis pouchetii* and *Pseudonitzschia delicatissima*	Human colon adenocarcinoma cancer line Caco-2	┴Proliferation;↑cytotoxicity	11–17 µg/mL	[166]
	2-trans-4-cis-7-cis-decatrienal (**111**)					
	2-trans-4-trans-7-cis-decatrienal (**112**)					
	2-trans,4-trans-heptadienal (**113**)	*Skeletonema marinoi*	Lung adenocarcinoma cell line A549, and colon COLO 205	↑Cytotoxicity; ┴cell cycle at either G1 or S Phase	10 µM	[167,168]
	2-trans,4-trans–octadienal (**114**)		Lung adenocarcinoma cell line A549	┴Cell cycle at either G1 or S Phase	5 µM	
Polysaccharide	Chrysolaminaran polysaccharide (**115**)	*Synedra acus*	Human colon cancer cell lines HTC-116 and DLD-1	┴Proliferation	54.5 and 47.7 µg/mL	[169,170]
Sulfated polysaccharide	Fucoidans (**116**)	*Sargassum hornery*, *Eclonia cava* and *Costaria costata*	Human skin melanoma cell line (SK-MEL-28) and human colon cancer cell line (DLD-1)	┴Cancer	100 μg/mL	[171,172,173]
			MDA-MB-231 cells	↑Apoptosis	820 μg/mL	[174,175]
			Human lung cancer cells (A549)	┴ERK1/2 pathway;┴Metastatic activity;┴PI3K/Akt/mTOR pathway	400 μg/mL	[176]
			Human hepatocellularcarcinoma cells (Huh7);HepG2 cells	┴Proliferation	2.0 and 4.0 mg/mL	[177,178,179,180]
		*Fucus evanescens*	C57Bl/6 mice	┴Growth of tumor	10 mg/kg	[174]
Anionic polysaccharide	Alginic acid (**117**)	*Sargassum wightii*	H22 tumor-bearing mice	┴Growth of tumor	Not specified	[41]
Polysaccharide	Laminarin (**118**)	*Eisenia bicyclis*	ES2 (ovarian clear cell carcinoma cells); OV90 (papillary serous adenocarcinoma cells) cell lines	┴Proliferation;↑apoptosis;┴cell cycle at subG1 Phase	2 mg/mL	[181,182]
			JB6 Cl41 (normal mouse epidermal cells); SK-MEL-28 (human malignant melanoma) cells	┴Cancer	Not specified	[183,184]
			Human colon cancer cell lines, such as HCT-116, HT-29, and DLD-1	↑Cytotoxicity	200 μg/mL	[182,185,186,187,188]
			Human colon carcinoma cells (LoVo)	↑Apoptosis	Not specified	[189]
			Human colon cancer cell line (HT-29)	↑Apoptosis, ┴cell cycle at subG1 and G2-M Phase	5 mg/mL	[190,191,192]
Carotenoids	Violaxanthin (**119**)	*Dunaliella tertiolecta*	MCF-7 cancer cell line	↑Apoptosis;↑cytotoxicity	20 and 40 μg/mL	[193,194,195,196]
			L1210 (human MDR1 gene-transfected mouse lymphoma cells); MDA-MB-231 (human breast cancer cells)	┴P-glycoprotein (P-gp) and MRP1	Not specified	[197]
			Human MDR1 gene-transfected mouse lymphoma; MCF-7 (human breast cancer cell)			[198]
	Neoxanthin (**120**)	*Tetraselmis suecica*	HeLa; A549 cancer cells	↑Cytotoxicity	Not specified	[199]
	Fucoxanthin (**121**)	*Undaria pinnatifida*	Human leukemia cell line (HL-60)	┴Proliferation;↑apoptosis;┴cell cycle at G0/G1 Phase or G2/M Phase	22.6 μM	[200,201,202,203,204,205]
	Siphonaxanthin (**122**)	*Codium fragile, Caulerpa lentillifera* and *Umbraulva japonica*	Human leukemia cell line (HL-60)	↑Apoptosis; ↑chromatin condensation;↓Bcl-2;↑caspase-3;↑GADD5α;↑DR5	10 μM	[206]
			Human umbilical vein endothelial cells (HUVECs)	┴Angiogenic effect;↓FGF-2;↓FGFR-1;↓EGR-1	2.5 μM	[207,208,209]
	Zeaxanthin (**123**)	*Porphyridium cruentum*, *Isochrysis galbana*, *Phaeodactylum tricornutum*, *Tetraselmis suecica* and *Nannochloropsis gaditana*	Human colon adenocarcinoma cell line (HT-29)	↑Cytotoxicity	10 μM	[210,211]
Xanthophyll carotenoids	Lutein (**124**)	*Porphyridium cruentum*, *Isochrysis galbana*, *Phaeodactylum tricornutum*, *Tetraselmis suecica* and *Nannochloropsi sgaditana*	Human colon adenocarcinoma cell line (HT-29)	↑Cytotoxicity	Not specified	[211]
Sterol	Stigmasterol (**125**)	*Navicula incerta*	Human liver cancer cell line (HepG2)	↑Cytotoxicity;┴proliferation;↑apoptosis;┴cell cycle at G0/G1 and G2/M Phase;↑caspase-8;↑caspase-9;↑Bax;↑p53;↓Bcl-2;↓XIAP	20 μM	[213,214]
Fatty alcohol ester	Nonyl 8-acetoxy-6-methyloctanoate (**126**)	*Phaeodactylum tricornutum*	Human promyelocytic leukemia cell line (HL-60), a human lung carcinoma cell line (A549) and a mouse melanoma cell line (B16F10).	↑Apoptosis;┴cell cycle at the sub G1 Phase	65.15 μM, 50μg/mL, not specified	[215]
Epimeric carotenoids	Dinochrome A and B (**127**, **128**)	*Peridinium bipes*	GOTO (neuroblastoma cells); OST (osteosarcoma cells) and HeLa cells	┴Proliferation;┴TPA-stimulated 32P-incorporation into the phosholipids of HeLa cells	5 μg/mL and 25 μg/mL	[216]
Porphyrin Phaeophytins	Porphyrinolactone (**129**)	*Cladophora fascicularis*	HeLa carcinoma cell line	┴Proliferation;┴activation of NF-κB	50 μM	[217]
	20-chlorinated (13^2^-S)-hydroxyphaeophytin A (**130**)					
	(132-S)-hydroxyphaeophytin A (**131**) and B (**132**)					
	(132-R)-hydroxyphaeophytin A (**133**) and B (**134**)					
Glycolipid	Nigricanosides A (**135**) and B (**136**) and methyl esters of nigricanosides A (**137**) and B (**138**)	*Avrainvillea nigricans*	Human breast cancer MCF-7 cells and human colon cancer HCT-116 cells	┴Proliferation, antimitotic activity, ↑tubulin polymerization within the cell	Not specified	[218]

Various symbols (↑, ↓ and ┴) indicate increase, decrease and inhibition in the obtained variables, respectively.

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
