# Peer review of "Marine Cyanobacteria and Microalgae Metabolites—A Rich Source of Potential Anticancer Drugs"

_marinedrugs, 2020, doi:10.3390/md18090476_

Round 1

Reviewer 1 Report

The review describes well compounds with anti tumor activities , as mentioned before the report could be shorter.

Author Response

The authors of this manuscript express their sincere thanks to the reviewer for the critical assessment of this work. The authors have acted upon the recommendations of the  reviewer which have resulted in a significant enhancement of the quality of this manuscript. The major modifications incorporated in the manuscript are highlighted using red color font. A “point-by-point” response to each and every comment is outlined below.

Comment:

The review describes well compounds with anti tumor activities , as mentioned before the report could be shorter.

Response:

We are deeply encouraged by the reviewer’s generous comments about the quality of our work. We have tried our best to reduce the length, but are also required to incorporate new text as per other reviewers’ comments.

Reviewer 2 Report

The manuscript 'Marine Cyanobacteria and Microalgae Metabolites - A rich Source of Potential Anticancer Drugs' deals with an interesting and important topic. An up to date review about cyanobacteria and mircoalgae origin natural products with potent anticancer activity would be helpful for the scientific community.

However, this manuscript in the present form is not ready to be published.

1.) The manuscript is written by several authors. Chapters 4–6 include and introduction part, respectively. All of these introduction are pointing out the importance of natural products and hihglighting their potencial anticancer activity. Similar repetitions are in the Introduction part. In my opinion, the compilation of this parts wort not sufficient during the manuscript preparation.

The introductions form the beginning of the chapters (i.e. lines 213-218; 278-292; 685-689) should be moved to Introduction. I suggest writing a new Introduction which is more concised and focused.

2.)  According to the title, the foci of this manuscript are cyanobacteria and microalgae. Dolabella auricularia, Ecteinascidi turbinate, Aplidium albicans are belonging to Animalia. What is the concept on which are these taxa not belonging to Cyanobacteria and Microalgae involved?

3.) There are several definitions given in the manuscript which are arguable. There are plenty of alkaloids in which nitrogen is within its ring structure (e.g. capsaicin). Not all terpene compounds are unsaturated and volatile (e.g. ursane). I would recommend to the authors to skip these definitions.

4.) In '2. Literature seach methodology' it is mentioned that PRISMA was followed. PRISMA is used for Systematic reviews and metaanalysis of clinical reviews. However, if the articles were selected following PRISMA (or any Extension of PRISMA), it would be great to see the grading of the articles, the inclusion and exclusion criteria, flow chart of the selection process.

5.) Please consider hedging in case of second sentence of Abstract.

6.) Give the compound number in the tables. It will make more easy to find the related structure.

7.) Check the species names. Be sure that genus name starts with capital letter (e.g. line 377. leptolyngbya sp.). Check the cancer cell line names and make the unified (e.g. LoVo in line 147 ↔ LOVO in line 675).

8.) Figure 14. is missing.

Author Response

The authors of this manuscript express their sincere thanks to the reviewer for the critical assessment of this work. The authors have acted upon the recommendations of the  reviewer which have resulted in a significant enhancement of the quality of this manuscript. The major modifications incorporated in the manuscript are highlighted using red color font. A “point-by-point” response to each and every comment is outlined below.

General comments:

The manuscript 'Marine Cyanobacteria and Microalgae Metabolites - A rich Source of Potential Anticancer Drugs' deals with an interesting and important topic. An up to date review about cyanobacteria and mircoalgae origin natural products with potent anticancer activity would be helpful for the scientific community.

However, this manuscript in the present form is not ready to be published.

Response:

We sincerely thank the reviewer for his/her expertise as well as time and effort for reviewing our manuscript with constructive suggestions. As described below, we have revised our manuscript as per the reviewer’s worthy comments

Specific comments:

Comment 1:

The manuscript is written by several authors. Chapters 4–6 include and introduction part, respectively. All of these introduction are pointing out the importance of natural products and hihglighting their potencial anticancer activity. Similar repetitions are in the Introduction part. In my opinion, the compilation of this parts wort not sufficient during the manuscript preparation.

The introductions form the beginning of the chapters (i.e. lines 213-218; 278-292; 685-689) should be moved to Introduction. I suggest writing a new Introduction which is more concised and focused.

Response:

We completely agree with the reviewer’s suggestion. Hence, we have modified the introduction section (page 2, lines 46-57; page 2, 70-73; and page 2, lines 84-87).

Comment 2:

According to the title, the foci of this manuscript are cyanobacteria and microalgae. Dolabella auricularia, Ecteinascidi turbinate, Aplidium albicans are belonging to Animalia. What is the concept on which are these taxa not belonging to Cyanobacteria and Microalgae involved?

Response:

Several metabolites were first identified in Dolabella auricularia, Ecteinascidi turbinate, and Aplidium albicans which belongs to Animalia. But latter on these metabolites were also extracted from cyanobacteria and microalgae. We completely agree with this fact that this manuscript deals with only cyanobacteria and microalgae. Accordingly, we have deleted text on Dolabella auricularia, Ecteinascidi turbinate, and Aplidium albicans (page 5, last paragraph and page 6, second paragraph).

Comment 3:

There are several definitions given in the manuscript which are arguable. There are plenty of alkaloids in which nitrogen is within its ring structure (e.g. capsaicin). Not all terpene compounds are unsaturated and volatile (e.g. ursane). I would recommend to the authors to skip these definitions.

Response:

This is an excellent point. We have modified the definitions as suggested (page 3, lines 143 and 144; page 4, lines 171-173).

Comment 4:

In '2. Literature seach methodology' it is mentioned that PRISMA was followed. PRISMA is used for Systematic reviews and metaanalysis of clinical reviews. However, if the articles were selected following PRISMA (or any Extension of PRISMA), it would be great to see the grading of the articles, the inclusion and exclusion criteria, flow chart of the selection process.

Response:

We greatly appreciate this thought-provoking comment. We have revised the methodology section to clearly incicate inclusion and exclusion criteria (page 3, line 132; page 3, lines 132-134; and page 3, lines 136 and 137). We have not used any grading and excluded additional information to conserve space as per the comments from other reviewers.

Comment 5:

Please consider hedging in case of second sentence of Abstract.

Response:

We have deleted this sentence from the abstract.

Comment 6:

Give the compound number in the tables. It will make more easy to find the related structure.

Response:

We believe the reviewer has made an excellent point. We have included the compound numbers in the tables 1 and 2.

Comment 7:

Check the species names. Be sure that genus name starts with capital letter (e.g. line 377. leptolyngbya sp.). Check the cancer cell line names and make the unified (e.g. LoVo in line 147 ↔ LOVO in line 675).

Response:

We are grateful to the reviewer for these close observations. We have thoroughly checked the manuscript and made the corrections as suggested (page 19, line 371 and page 35, line 669).

Comment 8:

Figure 14. is missing.

Response:

We sincerely apologize for this inadvertent error. Figure 14 has been included in our revised manuscript (page 32).

Additionally,

  1. The reference list has been modified as we have added several new references and deleted a few. Special attention is given to conform to the order of references and bibliographic style of the journal.
  2. The entire manuscript has been thoroughly checked and edited to ensure uniform style, organization, and quality.

On behalf of my co-authors, I once again express my sincere thanks to the erudite reviewer for the valuable suggestions and constructive input to improve the quality of our manuscript.

Reviewer 3 Report

Very interesting review. I suggest some minor revisions only, mostly regarding English language and two recent reviews published on the same subject (to be included in the reference list). 

Minor revisions are detailed below:

Abstract, change centered towards into focused on

Introduction, line 58, unexplored field of research

line 60 are expensive, tedious

line 68, change This endurance of life as we know, into: This extreme biodiversity encompasses a heterogeneous array of micro- and macroorganisms.

line 69, fungi

line 101, tio?18-20 (remove tio)

the order of references 42 and 43 should be inverted

line 334, remove (figure 2)

Figure 14 is not visible

The references must be checked for editing, i.e. references 122, 123 and 164 report species names not in italics

reference 160, Lyngbya majuscula

reference 163, Nostoc

reference 213, Phaeodactylum tricornutum

I suggest to include in the reference list Raposo et al (2013)

Health applications of bioactive compounds from marine microalgae (by
Maria Filomena de Jesus Raposo, RuiManuel Santos Costa de Morais, AlcinaMaria Miranda Bernardo deMorais, Life Sciences, 93: 479–486)

and also 

Microalgae metabolites: A rich source for food and medicine
(by Ramaraj Sathasivam a, Ramalingam Radhakrishnan b,⇑, Abeer Hashem c, Elsayed F. Abd_Allah, Saudi Journal of Biological Sciences 26 (2019) 709–722)

Anticancer Compounds Derived fromMarine Diatoms (by Hanaa Ali Hussein and Mohd Azmuddin Abdullah, Marine drugs, 2020, 18, 356)

Microalgal Derivatives as Potential Nutraceutical and Food Supplements for Human Health: A Focus on Cancer Prevention and Interception (By Galasso et al., 2019, Nutrients, 11, 1226)

Author Response

The authors of this manuscript express their sincere thanks to the reviewer for the critical assessment of this work. The authors have acted upon the recommendations of the  reviewer which have resulted in a significant enhancement of the quality of this manuscript. The major modifications incorporated in the manuscript are highlighted using red color font. A “point-by-point” response to each and every comment is outlined below.

General comments:

Very interesting review. I suggest some minor revisions only, mostly regarding English language and two recent reviews published on the same subject (to be included in the reference list).

Response:

We are deeply encouraged by the reviewer’s generous comments about the quality of our work. We sincerely apologize for our oversight and now have included in our manuscript the important reviews suggested by the reviewer (see below). We have also extensively edited our manuscript to limit typographical and grammatical errors.

Specific comments:

Comment 1:

Abstract, change centered towards into focused on

Response:

We have made the correction as suggested (page 1, lines 31 and 32).

Comment 2:

Introduction, line 58, unexplored field of research

Response:

The necessary corrections has been made as indicated (page 2, line 69).

Comment 3:

line 60 are expensive, tedious

Response:

We have made the correction as suggested (page 2, line 74).

Comment 4:

line 68, change This endurance of life as we know, into: This extreme biodiversity encompasses a heterogeneous array of micro- and macroorganisms.

Response:

We have replaced the sentence with the suggested one (page 2, line 81).

Comment 5:

line 69, fungi

Response:

The suggested correction has been made (page 2, line 82).

Comment 6:

line 101, tio?18-20 (remove tio)

Response:

We have made the correction as suggested (page 3, line 119).

Comment 7:

the order of references 42 and 43 should be inverted

Response:

We are sorry for this error. In our revised manuscript,  reference 43 have been replaced with reference 52 and reference  43 has been renumbered to reference 58 in order to maintain chronological order of the references.

Comment 8:

line 334, remove (figure 2)

Response:

The necessary correction has been made (page 16, line 328).

Comment 9:

Figure 14 is not visible

Response:

We sincerely apologize for this inadvertent error. Figure 14 has been included (page 32).

Comment 10:

The references must be checked for editing, i.e. references 122, 123 and 164 report species names not in italics

Response:

We have made the corrections in species name which have been written in italics in references 121, 122 and 163 (pages 51 and 53).

Comment 11:

reference 160, Lyngbya majuscula

reference 163, Nostoc

reference 213, Phaeodactylum tricornutum

Response:

Similar corrections have been made as per the suggestions in references 159, 162 and 215 (pages 53 and 55).

Comment 12:

I suggest to include in the reference list Raposo et al (2013)

Health applications of bioactive compounds from marine microalgae (by

Maria Filomena de Jesus Raposo, RuiManuel Santos Costa de Morais, AlcinaMaria Miranda Bernardo deMorais, Life Sciences, 93: 479–486)

and also

Microalgae metabolites: A rich source for food and medicine

(by Ramaraj Sathasivam a, Ramalingam Radhakrishnan b,, Abeer Hashem c, Elsayed F. Abd_Allah, Saudi Journal of Biological Sciences 26 (2019) 709–722)

Anticancer Compounds Derived fromMarine Diatoms (by Hanaa Ali Hussein and Mohd Azmuddin Abdullah, Marine drugs, 2020, 18, 356)

Microalgal Derivatives as Potential Nutraceutical and Food Supplements for Human Health: A Focus on Cancer Prevention and Interception (By Galasso et al., 2019, Nutrients, 11, 1226)

Response:

We are indebted to the reviewer for suggesting these excellent publications. We have included all 4 references in the revised manuscript as reference no. 194, 195, 212 and 214.

Additionally,

  1. The reference list has been modified as we have added several new references and deleted a few. Special attention is given to conform to the order of references and bibliographic style of the journal.
  2. The entire manuscript has been thoroughly checked and edited to ensure uniform style, organization, and quality.

On behalf of my co-authors, I once again express my sincere thanks to the erudite reviewer for the valuable suggestions and constructive input to improve the quality of our manuscript.

Round 2

Reviewer 2 Report

Thank you for the modifications. With these additions and modifications this paper has been improved and it will be a significant and useful review for the scientific community.

My last and only comment, which I still have, is related to PRISMA method. Application of PRISMA requires more details about review process itself. These data are the essence of PRISMA (e.g. database version, literature screen date, flow-chart, and grading of literature). I ask you kindly to attach the flow diagram (http://prisma-statement.org/PRISMAStatement/FlowDiagram.aspx) and checklist (http://prisma-statement.org/PRISMAStatement/Checklist.aspx). In case you do not have these attachments, please remove the last sentence from ‘2. Literature search methodology’ part (lines 137-138).

Author Response

We express our sincere thanks to the reviewer for re-evaluation of our manuscript. We are pleased to learn that our manuscript has been improved significantly. 

We once again revised our manuscript as per the comment. We have removed the last line of section 2: Literature search methodology (page 3).

We hope our manuscript is now acceptable for publication.

With best regards.

This manuscript is a resubmission of an earlier submission. The following is a list of the peer review reports and author responses from that submission.